



# Multi-compartment kinetic-allometric model of radionuclide bioaccumulation in marine fish

Roman Bezhenar[1], Kyeong Ok Kim[2], Vladimir Maderich[1], Govert de With[3], and Kyung Tae Jung[4]

[1]Institute of Mathematical Machine and System Problems, Glushkov av., 42, Kyiv 03187, Ukraine
[2]Korea Institute of Ocean Science and Technology, Metropolitan city Busan, Republic of Korea
[3]NRG, Utrechtseweg 310, 6800 ES Arnhem, the Netherlands
[4]Oceanic Consulting & Trading, 403 Munhwa-building, 90 Yangpyong-ro, Seoul, Republic of Korea

**Correspondence:** Vladimir Maderich (vladmad@gmail.com)

**Abstract.** A model of the radionuclide accumulation in fish taking into account the contribution of different tissues and allometry is presented. The basic model assumptions are as follows: (i) A fish organism is represented by several compartments in which radionuclides are homogeneously distributed; (ii) The compartments correspond to three groups of organs/tissues: muscle, bones and organs (kidney, liver, gonads, etc.) differing in metabolic function; (iii) Two input compartments include

gills absorbing contamination from water and digestive tract through which contaminated food is absorbed; (iv) The absorbed radionuclide is redistributed between organs/tissues according to their metabolic functions; (v) The elimination of assimilated elements from each group of organs/tissues differs, reflecting differences in specific tissues/organs in which elements were accumulated; and (vi) The food and water uptake rates, elimination rate and growth rate depend on the metabolic rate, which is scaled by fish mass to the $3/4$ power. The analytical solutions of the system of model equations describing dynamics of

the assimilation and elimination of $^{134}$Cs, $^{57}$Co, $^{60}$Co, $^{54}$Mn and $^{65}$Zn, which are preferably accumulated in different tissues, exhibited good agreement with the laboratory experiments. The developed multi-compartment kinetic-allometric model was embedded into the compartment model POSEIDON-R, which describes transport of radionuclides in water, accumulation in the sediment, and transfer of radionuclides through the pelagic and benthic food webs. The POSEIDON-R model was applied for the simulation of the transport and fate of $^{60}$Co and $^{54}$Mn routinely released from Forsmark NPP located on the Baltic

Sea coast of Sweden and for calculation of $^{90}$Sr concentration in fish after the accident at Fukushima Dai-ichi NPP. Predicted concentrations of radionuclides in fish agree with the measurements much better than predicted using standard whole-body model and target tissue model. The model with the defined generic parameters could be used in different marine environments without calibration based on *a posteriori* information, which is important for emergency decision support systems

## 1   Introduction

Accumulation of radionuclides in marine organisms is a complicated process that is governed by uptake of radionuclides from water, sediment and food and by depuration. In turn, these processes depend on the chemical properties of elements, their roles in metabolic processes, the positions of organisms in the food web and marine environmental parameters. In the case of chronic exposure, the radiological assessment models often assumed an equilibrium approach (Carvalho, 2018), in which





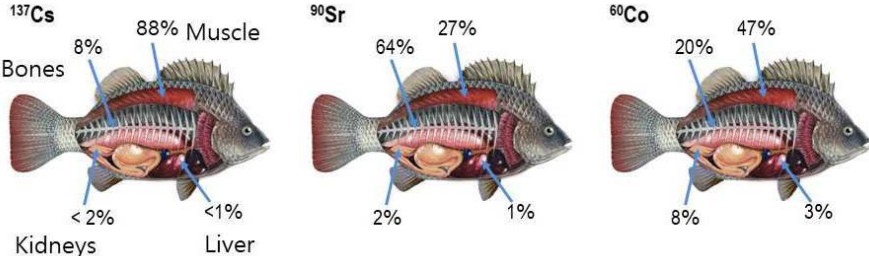

**Figure 1.** Distribution of accumulated $^{137}$Cs, $^{90}$Sr and $^{60}$Co in muscle, bone, liver and kidneys according to previously reported data (Yankovich et al., 2010).

concentration in the organism relates to the concentration in water using a biological accumulation factor ($BAF$). However, to describe highly time dependent transfer processes resulting from accidental releases, dynamic models for the uptake and retention of activity in marine organisms are necessary (Vives i Batlle et al., 2016). According to Takata et al. (2019), the effective recession times of post-Fukushima Dai-ichi Nuclear Power Plant (FDNPP) accident disequilibrium of $^{137}$Cs in biota ranged from 100 to 1,100 days. The most commonly used bioaccumulation models are the whole-body models, where the organism is represented as a single box in which contamination is evenly distributed (e.g. Fowler and Fisher, 2004; Tateda et al., 2013; Vives i Batlle et al., 2016). However, the distribution of radionuclides in organisms, and in particular in fish, is non-uniform. For example, the highest concentration of radiocaesium in fish is observed in the muscle, while the highest concentrations of the actinides, plutonium and americium, are measured in specific organs (Coughtrey and Thorne, 1983). Moreover, Vives i Batlle (2012) noted that elimination of activity from organisms occurred with different rates that can be interpreted as elimination from different tissues/organs with different metabolism. In a first approximation, this is used in the "target tissue" approach (Heling et al., 2002; Maderich et al., 2014a,b; Bezhenar et al., 2016), where radionuclides are grouped into several classes depending on the type of tissues in which a specific radionuclide accumulates preferentially (target tissue). However, the contribution of other tissues with greater mass than the mass of the target tissue can be commensurate with the contribution of the target tissue to the amount of radioactivity in the body. This is observed in Fig. 1, which is built from data (Yankovich et al., 2010) where the accumulated activity of $^{90}$Sr in muscle is not negligible in comparison with accumulated activity in bones, whereas the accumulated activity of $^{60}$Co is redistributed between muscle, bone and organs.

A more general approach to the description of the radionuclide accumulation in the tissues of fish is using the physiologically based pharmacokinetic (PBPK) models (Barron et al., 1990; Thomann et al., 1997; Garnier-Laplace et al., 2000; Otero-Muras et al., 2010). In the PBPK models, the fish organism is represented as three groups of compartments: absorption compartments simulating uptake of contaminants, distribution compartments simulating tissues and organs, and excretion compartments. The exchange of contaminants between compartments is limited by blood flux perfusing compartments. However, these models require a significant number of parameters depending on elements, fish species and marine environments. They must be determined from the laboratory experiments (Thomann et al., 1997) or by the optimization procedures (Otero-Muras et al., 2010).





Notice that PBPK fish models do not yet include scaling (allometric) relationships between metabolic rates and organism mass (West et al., 1997; Higley and Bytwerk, 2007; Vives i Batlle et al., 2007; Beresford et al., 2016). Therefore, there is a need

to develop a generic model of intermediate complexity between the one-compartment model and the PBPK model taking into account (i) the heterogeneity of the distribution of contamination in fish tissues and (ii) the allometric relationships between metabolic rates and organism mass. Such a model can be used for accidental release simulations without local calibration, which is a complicated task in the circumstances of the accident.

In this paper, a new approach for predicting radionuclide accumulation in fish taking into account the contributions of

different tissues and allometry is presented. The paper is organized as follows. The model is described in Section 2. The comparison with laboratory experiments is given in Section 3. The results of simulation of several radionuclides in the marine environment for regular and accidental releases are described in Section 4. The conclusions are presented in Section 5.

## 2 Model

### 2.1 Model equations

Here, a simple multi-compartmental model to simulate kinetics of radionuclides in the fish is described. The basic assumptions are as follows: (i) a fish organism is represented by several compartments in which radionuclides are homogeneously distributed; (ii) the compartments correspond to three groups of organs/tissues differing in metabolic function: flesh, bones and organs (kidney, liver, gonads, etc.); (iii) two input compartments include gills which absorb contamination from water and digestive tract through which contaminated food is absorbed; (iv) the absorbed radionuclide is redistributed between

organs/tissues according to their metabolic functions; (v) the elimination of assimilated elements from each group of organs/tissues differs, reflecting differences in the specific tissues/organs in which elements were accumulated; (vi) the food and water uptake rates, elimination rate and growth rate depend on the metabolic rate, which is scaled by fish mass to the $3/4$ power following general theory (West et al., 1997) describing transport of essential materials through space-filling fractal networks of branching tubes in organism.

The equation for concentration of radionuclide [Bq kg$^{-1}$ wet weight (WW)] in the gill compartment ($i = 1$) is written as

$$\frac{dm_1 C_1}{dt} = K_w m C_w - k_1 m_1 C_1 - \lambda_1 m_1 C_1. \tag{1}$$

The equation for concentration of unabsorbed radionuclide [Bq kg$^{-1}$ WW] in the digestive tract compartment ($i = 2$) is

$$\frac{dm_2 C_2}{dt} = K_f m C_f - k_2 m_2 C_2 - \lambda_2 m_2 C_2. \tag{2}$$

The equations for concentrations of radionuclide [Bq kg$^{-1}$ WW] in the muscle ($i = 3$), bones ($i = 4$) and organs ($i = 5$) are

$$\frac{dm_i C_i}{dt} = k_{2i} m_2 C_2 + k_{1i} m_1 C_1 - \lambda_i m_i C_i. \tag{3}$$

Here, $t$ is time; $m_i$ is the mass of the $i$-th tissues; $m$ [kg] is the total mass of fish; $C_w$ [Bq m$^{-3}$] is the concentration of radionuclide in water; $C_f$ [Bq kg$^{-1}$ WW] is the concentration of radionuclide in food; $K_w$ [(m$^3$(kg d)$^{-1}$] is a water uptake





rate per unit fresh weight of fish and $K_f$ [(kg(kg d)$^{-1}$] is a food uptake rate per unit fresh weight of fish; $\lambda_1$ is a loss rate from gills to water [d$^{-1}$]; $k_1$, $k_{1i}$, $k_2$, and $k_{2i}$ [d$^{-1}$] are transfer rates between tissues, $\lambda_2$ [d$^{-1}$] is the egestion rate from the digestive

tract; $\lambda_i$ [d$^{-1}$] is the absorbed radionuclide elimination rate from different tissues/organs ($i = \overline{3,5}$).

The activity concentration in the food $C_f$ is expressed by the following equation, summing for a total of $n$ prey types

$$C_f = \sum_{j=0}^{n} C_{prey,j} P_j \frac{drw_{pred}}{drw_{prey,j}}, \tag{4}$$

where $C_{prey,j}$ is the activity concentration in prey of type $0 \le j \le n$, $P_j$ is preference for prey of type $j$, $drw_{pred}$ is the dry weight fraction of fish, and $drw_{prey,j}$ is the dry weight fraction of prey of type $j$. The mean whole-body concentration of

activity in the organism $C_{wb}$ and whole-body activity $A_{wb}$ [Bq] are calculated as

$$A_{wb} = mC_{wb} = \sum_{i=1}^{5} m_i C_i = m \sum_{i=1}^{5} \mu_i C_i, \tag{5}$$

where $\mu_i$ are weighting factors, ($\sum_{i=1}^{5} \mu_i = 1$).

Transfer rates $k_1$ and $k_2$ are related with tissue transfer rates $k_{1i}$ and $k_{2i}$ as

$$k_1 = \sum_{i=3}^{5} k_{1i}, \quad k_2 = \sum_{i=3}^{5} k_{2i}. \tag{6}$$

Summing eqns. (1) to (3) yields the equation for total concentration of activity in fish $C_{wb}$ as

$$\frac{dC_{wb}}{dt} = K_w C_w + K_f C_f - \lambda_1 \mu_1 C_1 - \lambda_2 \mu_2 C_2 - \sum_{i=3}^{5} \lambda_i \mu_i C_i - \lambda_g C_{wb}, \tag{7}$$

where $\lambda_g$ is the organism growth rate, defined as

$$\lambda_g = \frac{1}{m}\frac{dm}{dt}. \tag{8}$$

The growth dilution can be ignored in the model calculations when $\lambda_g \ll \lambda_i$. For short-lived radionuclides, $\lambda_i$ should be

corrected taking into account the physical decay. The assimilation efficiencies of elements from water $AE_w$ and food $AE_f$ (Pouil et al., 2018) can be introduced, assuming that uptake from water and food is equilibrated by loss to the water from gills and through the egestion. The corresponding relations are

$$AE_w = \frac{K_w C_w - \lambda_1 \mu_1 C_1}{K_w C_w}, \tag{9}$$

$$AE_f = \frac{K_f C_f - \lambda_2 \mu_2 C_2}{K_f C_f}. \tag{10}$$

Taking into account the relations (9)-(10) for constant $AE_w$ and $AE_f$, the equation (7) will be similar to the standard whole-body single compartment equation

$$\frac{dC_{wb}}{dt} = AE_w K_w C_w + AE_f K_f C_f - (\lambda_{wb} + \lambda_g)C_{wb}, \tag{11}$$





**Table 1.** Parameters in allometric relations, standard deviation $STD$ of parameters and number of measurements $N$.

| Constant | Value | $N$ | $STD$ | Data source |
|---|---|---|---|---|
| $\alpha_w$ | 0.08 | 1 | - | Mathews et al. (2008) |
| $\alpha_f$ | 0.012 | 7 | 0.005 | Alava and Gobas (2016) |
| $\alpha_g$ | 0.0012 | 7 | 0.0003 | Alava and Gobas (2016) |
| $\alpha_1$ | 0.8 | - | - | This study |
| $\alpha_2$ | 0.75 | 7 | 0.18 | Andersen (1984); Pouil et al. (2017) |
| $\alpha_3$ | 0.007 | 8 | 0.0017 | Jeffree et al. (2006); Mathews and Fisher (2008); Mathews et al. (2008) |
| $\alpha_4$ | 0.001 | - | - | Heling et al. (2002) |
| $\alpha_5$ | 0.0275 | 1 | - | Rouleau et al. (1995) |

if elimination terms in (7) are replaced by a single term $\lambda_{wb}C_{wb}$, assuming that $\lambda_{wb}$ is a single whole-body elimination rate (Fowler and Fisher, 2004).

The food and water uptake rates, elimination rate and growth rate depend on the metabolic rate, which in turn is known to scale by the organism mass. Here, we employed quarter-power scaling for uptake, elimination and growth rates derived from general theory(West et al., 1997)

$$K_w(m) = \alpha_w m^{-1/4}, \quad K_f(m) = \alpha_f m^{-1/4}, \quad \lambda_g(m) = \alpha_g m^{-1/4}, \quad \lambda_i(m) = \alpha_i m^{-1/4}, \tag{12}$$

where $\alpha_w, \alpha_f, \alpha_g, \alpha_i$ $(i = \overline{1,5})$ are constants. These parameters can also depend on temperature, salinity and fish age (e.g. 110    Belharet et al., 2019; Heling and Bezhenar, 2009). Notice that a number of laboratory experiments (Thomas and Fisher, 2010) showed that temperature exerts no major influence on uptake and elimination, whereas the effect of salinity varies for elements (Heling and Bezhenar, 2009; Jeffree et al, 2017). Here, we did not analyze these factors requiring separate consideration. The values of constants $\alpha_w, \alpha_f, \alpha_g, \alpha_i$ $(i = \overline{1,5})$ estimated from laboratory experiments and marine data are provided in Table 1, whereas values of weighting factors $\mu_1$=0.01, $\mu_2$=0.01, $\mu_3$=0.78,$\mu_4$=0.12, $\mu_5$=0.08, were chosen using estimates from 115    Yankovich et al. (2010). The model parameters for characteristic values of the fish mass are given in Table S1 (Supplementary Material).



## 2.2 Kinetics in equilibrium state

The model parameters can be estimated using measurement data and applying the kinetic equations under equilibrium conditions. Equations (1) - (3) rewritten for radionuclide concentrations in the equilibrium state are

$$\mu_1 C_1 = \frac{K_w C_w}{k_1 + \lambda_1 + \lambda_g}, \tag{13}$$

$$\mu_2 C_2 = \frac{K_f C_f}{k_2 + \lambda_2 + \lambda_g}, \tag{14}$$

$$k_{2i} \frac{K_f C_f}{k_2 + \lambda_2 + \lambda_g} + k_{1i} \frac{K_w C_w}{k_1 + \lambda_1 + \lambda_g} = \lambda_i \mu_i C_i + \lambda_g \mu_i C_i. \tag{15}$$

Then, using (13)-(14), the assimilation efficiencies of elements from water $AE_w$ and food $AE_f$ are rewritten as

$$AE_w = \frac{k_1 + \lambda_g}{k_1 + \lambda_1 + \lambda_g}, \quad AE_f = \frac{k_2 + \lambda_g}{k_2 + \lambda_2 + \lambda_g} \tag{16}$$

When $\lambda_1 \gg \lambda g$, $\lambda_2 \gg \lambda g$, we determine from (16) and (12), approximately,

$$k_1 = \frac{AE_w \lambda_1}{1 - AE_w} = \frac{AE_w \alpha_1}{(1 - AE_w)m^{1/4}}, \quad k_2 = \frac{AE_f \lambda_2}{1 - AE_f} = \frac{AE_f \alpha_2}{(1 - AE_f)m^{1/4}}. \tag{17}$$

The equations under (17) is used to relate kinetic coefficients of the model with experimentally determined parameters $AE_w$, $AE_f$, $\lambda_1$ and $\lambda_2$. The equations under (15) can be rewritten as

$$AE_{fi} \alpha_f BAF_{food} + AE_{wi} \alpha_w = \mu_i (\alpha_i + \alpha_g) \frac{BAF_{wb}}{CR_i}, \tag{18}$$

where $AE_{wi}$ and $AE_{fi}$ are assimilation efficiencies for tissue $i$,

$$\sum_{i=3}^{5} \mu_i AE_{wi} \approx AE_w, \quad \sum_{i=3}^{5} \mu_i AE_{fi} \approx AE_f. \tag{19}$$

The assimilation efficiencies are expressed through kinetic coefficients as

$$AE_{wi} = \frac{k_{1i}}{k_1} AE_w, \quad AE_{fi} = \frac{k_{2i}}{k_2} AE_f. \tag{20}$$

The bioaccumulation factors for food $BAF_{food}$, for whole-body of fish $BAF_{wb}$ and body-to-tissue concentration ratio $CR_i$

are described as

$$BAF_{food} = \frac{C_{food}}{C_w}, \quad BAF_{wb} = \frac{C_{wb}}{C_w}, \quad CR_i = \frac{C_{wb}}{C_i}. \tag{21}$$

Values of $BAF$ for different radionuclides are available in IAEA (2004). Yankovich et al. (2010) provide $CR_i$ based on aggregate experimental data for marine fish. Assume that the kinetics of assimilation in fish tissues are similar for radionuclides absorbed from water and food, i.e.

$$\frac{AE_{wi}}{AE_w} = \frac{AE_{fi}}{AE_f}. \tag{22}$$





**Table 2.** The food assimilation efficiency $AE_f$ (Pouil et al., 2018), tissue assimilation efficiencies $AE_{fi}$ and $TTF$ for several elements.

| Element | $AE_f$ | $AE_{f3}$ | $AE_{f4}$ | $AE_{f5}$ | $TTF$ |
|---------|--------|-----------|-----------|-----------|-------|
| Cs | 0.76 | 0.88 | 0.08 | 0.65 | 1.3 |
| Sr | 0.29* | 0.21 | 0.64 | 0.51 | 0.71 |
| Co | 0.081 | 0.02 | 0.007 | 0.80 | 0.06 |
| Mn | 0.24 | 0.18 | 0.07 | 0.98 | 0.4 |
| Zn | 0.22 | 0.18 | 0.06 | 0.93 | 0.35 |
| Ag | 0.033 | 0.016 | 0.007 | 0.24 | 0.03 |
| Cu | 0.2 | 0.07 | 0.03 | 0.96 | 0.16 |
| Cd | 0.2 | 0.15 | 0.06 | 0.95 | 0.29 |
| Cr | 0.023 | 0.01 | 0.009 | 0.17 | 0.03 |

*This value was estimated using $BAF$ from IAEA (2004).

Notice that assimilation for some elements can be considered as route dependent (Reinfelder et al., 1999), and so (22) is only a first approximation. Inserting (22) into (18) yields

$$\frac{AE_{fi}}{AE_f}\left(AE_f\alpha_f BAF_{food} + AE_w\alpha_w\right) = BAF_{wb}\frac{\mu_i(\alpha_i + \alpha_g)}{CR_i},\tag{23}$$

Summing (23) for $i = 3, 4, 5$ yields

$$AE_f\alpha_f BAF_{food} + AE_w\alpha_w = BAF_{wb}\sum_{i=3}^{5}\frac{\mu_i(\alpha_i + \alpha_g)}{CR_i}.\tag{24}$$

Using (23) and (24), ratio $AE_{fi}/AE_f$ can be written as

$$\frac{AE_{fi}}{AE_f} = \mu_i(\alpha_i + \alpha_g)/\left(CR_i\sum_{i=3}^{5}\frac{\mu_i(\alpha_i + \alpha_g)}{CR_i}\right).\tag{25}$$

The values of kinetic coefficients $k_1$, $k_2$ and assimilation efficiencies for tissues $AE_{fi}$ were calculated from (17) and (25) using assimilation efficiencies from experimental data (Pouil et al., 2018). The values of $AE_{fi}$ for several radionuclides are given in Table 2. Equation (24) can be rearranged to express the ratio of $BAF_{wb}$ to $BAF_{food}$ taking into account dominance of dietary intake over water intake (Mathews and Fisher, 2009). This ratio is the trophic transfer factor ($TTF$), written as

$$TTF = AE_f\alpha_f/\sum_{i=3}^{5}\frac{\mu_i(\alpha_i + \alpha_g)}{CR_i}.\tag{26}$$

A $TTF > 1$ indicates possible biomagnification, and $TTF < 1$ indicates that biodiminution is likely. As follows from (26), the $TTF$ value does not depend on the mass of fish. The $TTF$ values calculated by using (26) are given in Table 2. Among the considered elements, only caesium ($TTF > 1$) may be biomagnified in the food chain, in agreement with Kasamatsu and Ishikawa (1997), where it was found that the $BAF$ of $^{137}$Cs increased with increasing trophic level. The concentrations of





other radionuclides in Table 2 decreased with the increase of trophic level ($TTF < 1$), which is consistent with the findings presented by Cardwell et al. (2013), where an inverse relationship was obtained between trophic levels and the concentration of inorganic metals in water chains.

Notice that values of $AE_f$, $AE_w$ and $BAF$ do not depend on the fish mass. The literature data reveal diverse relationships between fish mass and both radionuclide bioconcentration ($BCF$) and bioaccumulation ($BAF$) factors. In particular, data of laboratory experiments (Mathews et al., 2008) showed that there is no significant relationship between bioconcentration factor $BCF$ and fish size for most studied aqueous metals. The $BAF$ in larger and older fish of the same species can differ from smaller and younger fish due to the change of habitat and diet with age (e.g. Kasamatsu and Ishikawa, 1997; Ishikawa et al.,

1999; Kim et al., 2019) that does not contradict our findings.

## 3   Comparison with laboratory experiments

### 3.1   Depuration of radionuclides after pulse-like feeding

Retention of absorbed elements in fish after single feeding was often used to estimate $AE_f$ and depuration rate (e.g. Jeffree et al., 2006; Mathews and Fisher, 2008; Mathews et al., 2008; Pouil et al., 2017). According to Goldstein and Elwood (1971),

single feeding can be approximated by a delta function $\delta(t)$ at $t = 0$ as

$$K_f m C_f = A_f \delta(t), \tag{27}$$

where $A_f$ is the total amount of ingested activity. The solutions of equations (1)-(3) for activities $A_2 = m_2 C_2$ and $A_i = m_i C_i$ and for initial conditions $A_2 = A_i = 0$ are

$$\frac{A_2}{A_f} = \frac{m_2 C_2}{A_f} = \exp(-(k_2 + \lambda_2)t), \tag{28}$$

$$\frac{A_i}{A_f} = \frac{m_i C_i}{A_f} = \frac{k_{2i}}{k_2 + \lambda_2 - \lambda_i} \left[ \exp(-\lambda_i t) - \exp(-(k_2 + \lambda_2)t) \right]. \tag{29}$$

As follows from solutions (28)-(29), the decay of activity in the fish organisms includes a fast component with decay constant representing transfer of activity to the fish body and unabsorbed element egestion from the digestive tract, along with a slow component which is governed by elimination constants for $i$ tissues. These solutions are generalized solutions of the equations of the sequentially linked two-compartment model by Goldstein and Elwood (1971), whereas in the D-DAT model (Vives i

Batlle et al. 2008), an organism was represented by two boxes with parallel kinetics, also describing "slow" and "fast" exchange processes.

The solutions (28)-(29) can be compared with laboratory experiments in which depuration of metals from the fish after single feeding was studied. In the experiment by Mathews and Fisher (2008), the retention of several radioisotopes in juvenile sea bream (*Sparus auratus*) was considered. The average wet weight of the fish was 0.0001 kg. These fish were fed radiolabelled

*Artemia salina nauplii*. The fish were allowed to feed for one hour, after which metal retention was observed in clean water over a 15-day period. The solutions with parameters corresponding to the fish mass and metal $AE$ (Pouil et al., 2018) were





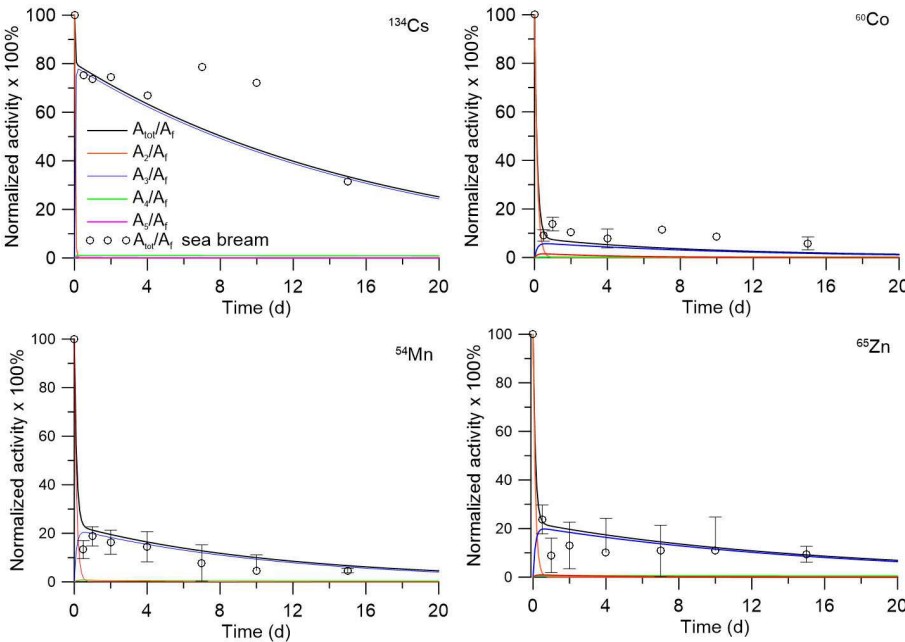

**Figure 2.** Retention of radionuclides in whole body and tissues of juvenile sea bream (*Sparus auratus*). The simulations are compared with whole body measurements by Mathews and Fisher (2008).

compared with experimental data for $^{134}$Cs, $^{60}$Co, $^{54}$Mn, and $^{65}$Zn in Fig. 2. As seen in Fig. 2, both model and experiments showed two phases (fast and slow) of radionuclide elimination. Most of the activity is contained in muscle; however, the concentrations of $^{60}$Co, $^{54}$Mn, and $^{65}$Zn in organs are much greater than in the muscle.

The solutions were also compared with laboratory experiments for predator fish (Mathews et al., 2008). In these experiments, the retention of several radioisotopes in sea bream (*Sparus auratus*), turbot (*Psetta maxima*) and spotted dogfish (*Scyliorhinus canicula*) was studied. Immature *S. auratus* (wet weight 0.012 kg), *P. maxima* (wet weight 0.027 kg), and *S. canicula* (wet weight 0.008 kg) were fed radiolabelled prey fish (juvenile *S. auratus*). After this feeding, fish were fed unlabelled prey fish for three weeks. In Fig. 3, the solutions (28)-(29) were compared with the laboratory experiments in which prey fish were
labelled by $^{134}$Cs, $^{57}$Co, $^{54}$Mn, and $^{65}$Zn. The solution (28)-(29) and experiment agreed, demonstrating general dependence of the depuration process on fish mass. Differences between experimental data for different species may be due to differences in anatomy and physiology, as discussed by Jeffree et al. (2006) for *P. maxima* and *S. canicula*. The model, unlike the situation for prey fish (Fig. 2), underestimates the total concentrations of $^{57}$Co and $^{54}$Mn in comparison with experiments, which is probably due to the neglect of other factors, except body weight, for the bioaccumulation kinetics.

**3.2    Bioconcentration of dissolved radionuclides from sea water**

Uptake and absorption in fish of elements from water were studied in several laboratory experiments (e.g. Jeffree et al., 2006; Mathews and Fisher, 2008; Mathews et al., 2008). The modelling of the absorption of elements can be used to estimate an




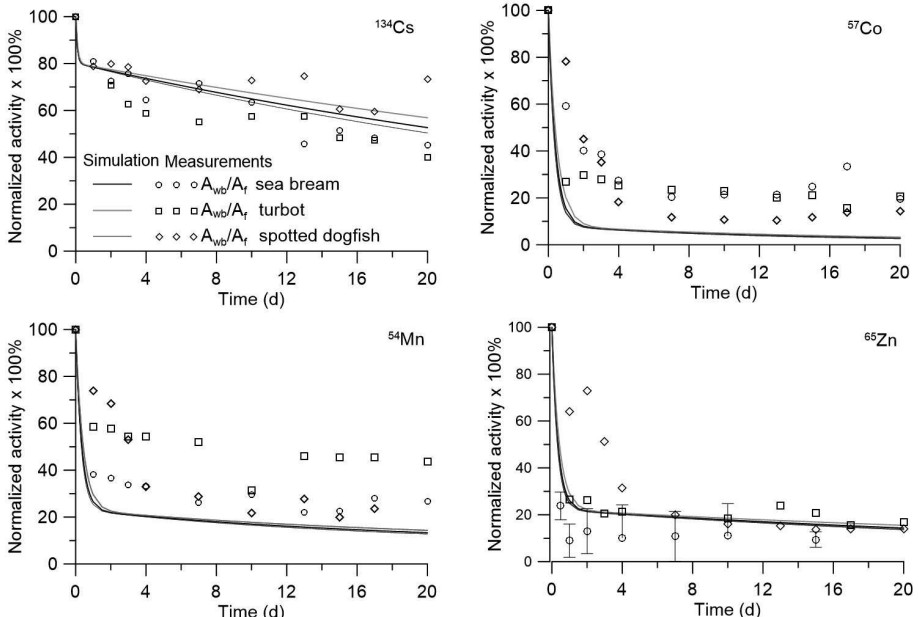

**Figure 3.** Retention of radionuclides in whole bodies of predator fish: sea bream (*Sparus auratus*), turbot (*Psetta maxima*) and spotted dogfish (*Scyliorhinus canicula*). The simulations are compared with whole body measurements by Mathews et al. (2008).

assimilation efficiency $AE_w$. An analytical solution of equations (1) and (3) with initial conditions $C_i = 0$ at $t = 0$ is written as

$$\frac{\mu_1 C_1}{C_w} = \frac{K_w}{k_1 + \lambda_1} \left[ 1 - \exp(-(k_1 + \lambda_1)t) \right], \tag{30}$$

$$\frac{\mu_i C_i}{C_w} = \frac{k_{1i} K_w}{k_1 + \lambda_1 - \lambda_i} \left[ 1 - \frac{k_1 + \lambda_1}{k_1 + \lambda_1 - \lambda_i} \exp(-\lambda_i t) + \frac{\lambda_i}{k_1 + \lambda_1 - \lambda_i} \exp(-(k_1 + \lambda_1)t) \right], \quad (i = \overline{3,5}) \tag{31}$$

These solutions were compared with laboratory experiments for prey fish (Mathews and Fisher, 2008) and for predator fish (Jeffree et al., 2006). In the experiment by Mathews and Fisher (2008), the uptake of several radioisotopes by juvenile *S. auratus* (wet weight 0.0002 kg) was studied during 25 days of exposure, whereas in experiments by Jeffree et al. (2006),

immature *P. maxima* (wet weight 0.0061 kg) and *S. canicula* (wet weight 0.0067 kg) were used for study during a 15-day period. The comparison of the analytical solution (30)-(31) with experimental data with respect to bioconcentration factor ($BCF = C_{wb}/C_w$ [$\mathrm{l\,kg^{-1}}$]) is presented in Fig. 4. The values of $AE_w$ were selected to approximate the experiment for small prey fish (Mathews, Fisher, 2008). They differ for different metals. For $^{134}$Cs, the value of $AE_w$ was 0.001, whereas for $^{57}$Co, $^{60}$Co, $^{54}$Mn, and $^{65}$Zn, these values were 0.005. This contrasted with $AE_f$, which is larger for $^{134}$Cs than for $^{57}$Co, $^{60}$Co,

$^{54}$Mn, and $^{65}$Zn (Pouil et al., 2017). The abovementioned values of $AE_w$ were used to calculate $BCF$ for larger predator fish in experiments by Jeffree et al. (2006). As seen in Fig. 4, values of $AE_w$ for considered elements are of the order $10^{-3}$, which is in agreement with most models. However, comparison with larger fish highlighted some differences between species of fish, as discussed by Jeffree et al. (2006), and differences between model and experiment for a constant value of $AE_w$. At the same

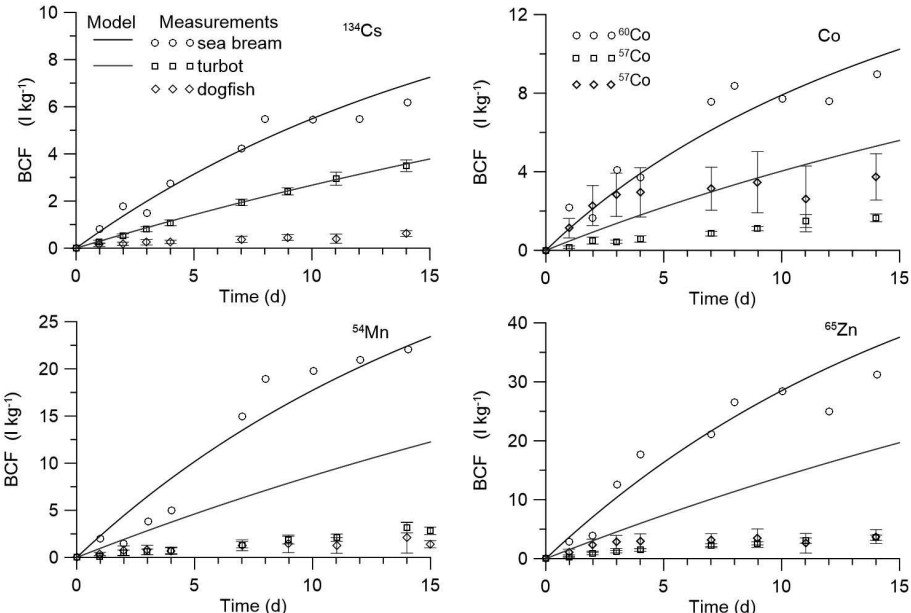

**Figure 4.** Simulated $BCF$ in marine fish during the period of exposition in water. The simulations are compared with isotope measurements by Mathews and Fisher (2008) in juvenile sea bream (*Sparus auratus*) and measurements by Jeffree et al. (2006) in turbot (*Psetta maxima*) and spotted dogfish (*Scyliorhinus canicula*).

time, it is known that dietary intake of metals dominates over water intake (Mathews and Fisher, 2009). Therefore, deviations
in values of $AE_w$ would not be significantly affected by the full uptake of elements from the marine environment.

The parameter $\alpha_1$ can be estimated from the relations (13) and (16) according to the experimental data for equilibrium conditions. An average value of gill $BCF_1$ is approximately $10 \, \mathrm{l \, kg^{-1}}$ for $^{58}$Co, $^{54}$Mn, $^{134}$Cs and $^{65}$Zn (Jefferies and Heweit,1971; Pentreath, 1973). Then for $AE_w$=0.001 we obtained $\alpha_1$=0.8. With the selected value of $\alpha_1$, the process of adaptation of the gill tissue to changes in the concentration of radioactivity in water is much faster than for other tissues of the fish. In addition,
as follows from the solution (30)-(31) at $\lambda_1 \gg \lambda_i$, the contribution of gill contamination to the whole-body contamination is small.

### 3.3 Simplification of the model based on the results of analytical solutions and laboratory experiments

Comparison of the model with laboratory experiments on the retention of absorbed elements in fish after single feeding has shown the importance of including in the model of the digestive tract compartment describing highly non-equilibrium trans-
fer dynamics. However, for modelling of food uptake in marine environment with multiple feedings the simple equilibrium assumption (14) can be used. At the same time, the analytical solution describing the bioconcentration due to the uptake and absorption in fish of elements from water, as well as the results of the laboratory experiment, show that the contribution to the gills (13) is negligible. Therefore, for modelling of uptake from water the equilibrium assumption can also be used as it shown





above. The corresponding simplified equations for muscle, bone and organs can be rewritten as

$$\mu_i \frac{dC_i}{dt} = AE_{fi}K_fC_f + AE_{wi}K_wC_w - \lambda_i\mu_iC_i - \lambda_g\mu_iC_i. \tag{32}$$

The uncertainty in calculations using equations (32) arises due to (i) limited experimental data to define allometry constants $\alpha_f$, $\alpha_w$, $\alpha_3$, $\alpha_4$, $\alpha_5$; (ii) large intervals in the values of known $AE_f$ coefficients; (iii) unknown $AE_f$ values for many radionuclides; (iv) lack of experimental data about $AE_w$; (v) limited experimental data for whole-body to tissue concentration ratios $CR_i$ in the marine fish. Therefore, a sensitivity analysis is necessary to estimate uncertainty of the simulation results. We estimated
the effects of variations in above parameters in the equation (24) on the value of $BAF_{wb}$ in equilibrium state. The simple local sensitivity analysis and One-At-a-Time method (Pianosi et al., 2016) was used. The sensitivity of model output was estimated using a sensitivity index ($SI$) calculated following Hamby (1994)as

$$SI = \frac{D_{max} - D_{min}}{D_{max}}, \tag{33}$$

where $D_{max}$ and $D_{min}$ are the outputs corresponded to maximal and minimal input parameter values, respectively. Similarly
to Bezhenar et al. (2016), the range for every parameter was defined as follows: minimum value was set to half the reference value and maximum value was set to twice the reference value. Calculated $SI$ for three radionuclides, which are preferably accumulated in different tissues: $^{137}$Cs, $^{90}$Sr, and $^{60}$Co, are given in Table. S2.

The results of sensitivity study suggest that model results are most sensitive to variations of $AE_f$ and $\alpha_f$ for $^{137}$Cs and $^{60}$Co, whereas for$^{60}$Co they are almost not sensitive to variations of $AE_w$ and $\alpha_w$. Note that model results are also sensitive to the
variations of parameters related to tissues where radionuclide is mainly accumulated: $\alpha_3$ and $CR_3$ for $^{137}$Cs, $\alpha_5$ and $CR_5$ for $^{60}$Co. For $^{90}$Sr the model results are moderately sensitive to the variations of most above considered parameters.

## 4  Model applications

### 4.1  Modified compartment POSEIDON-R model

In order to predict the accumulation of radionuclides in fish in the marine environment using the multi-compartment kinetic-
allometric (MCKA) model described above, it is necessary to calculate changes in concentration in water and in bottom sediments and to calculate the transport of radionuclides through food chains. The POSEIDON-R compartment model (Lepicard et al., 2004; Maderich et al., 2014a,b; 2018b; Bezhenar et al., 2016) can be used to simulate the marine environment as a system of 3D compartments for the water column, bottom sediment, and food web. The water column compartment is vertically subdivided into layers. The suspended matter settles in the water column. The bottom sediment compartment is divided into three
layers (Fig. S1). The downward burial processes operate in all three sediment layers. Maderich et al. (2018b) described the POSEIDON-R compartment model in detail. A food web model that includes pelagic and benthic food chains is implemented within the POSEIDON-R compartment model (Bezhenar et al., 2016). In the food web model, marine organisms are grouped into classes according to trophic level and species type (Fig. S2). The food chains differ between the pelagic zone and the benthic zone. Pelagic organisms comprise primary producers (phytoplankton) and consumers (zooplankton, non-piscivorous





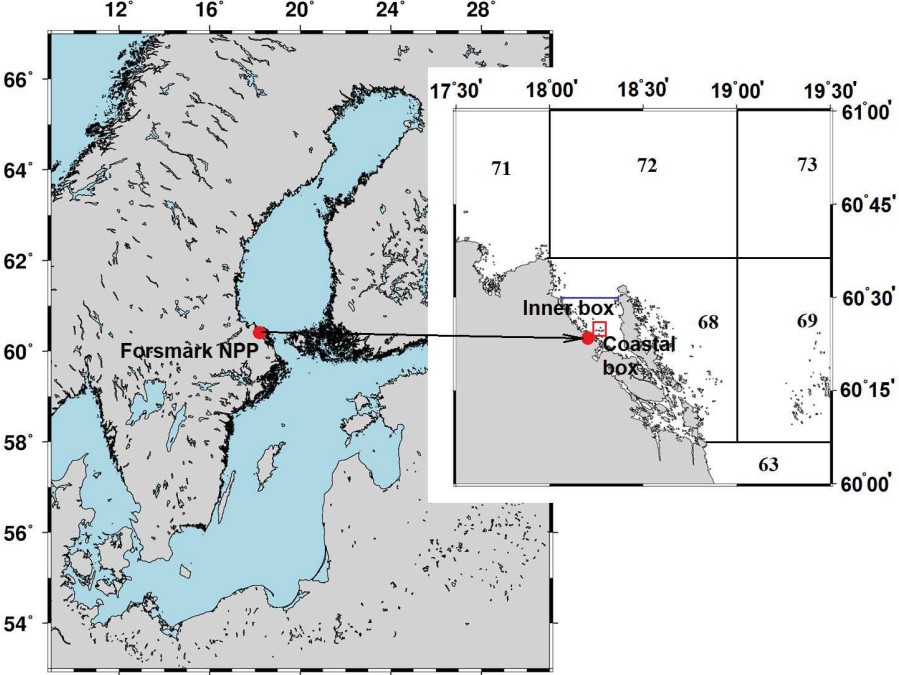

**Figure 5.** Compartment system around the Forsmark NPP. The numbers denote regional boxes in the box system of the Baltic Sea (Bezhenar et al., 2016). An additional "Inner box" is separated from regional box 68 by the blue line. The coastal box (red rectangle) surrounds the area, where cooling water from NPP is discharged.

(forage) fish, and piscivorous fish). In the benthic food chain, radionuclides are transferred from algae and contaminated bottom sediments to deposit-feeding invertebrates, demersal fish, and benthic predators. Bottom sediments include both organic and inorganic components. Radioactivity is assumed to be assimilated by benthic organisms from the organic components of the bottom deposits. Other food web components are crustaceans (detritus feeders), molluscs (filter feeders), and coastal predators, which feed throughout the water column in shallow coastal waters. All organisms take in radionuclides both via the food web

and directly from the water. Details of the transfer of radiocaesium through the marine food web are presented by Bezhenar et al. (2016) and Maderich et al (2018b).

## 4.2  Release of radionuclides during normal operation of the Forsmark nuclear power plant

This section presents the simulation results of $^{60}$Co and $^{54}$Mn routine release into the marine environment from Forsmark NPP, located on the Baltic Sea coast of Sweden. The compartment POSEIDON-R model with embedded food web MCKA model,

one-compartment model and bioaccumulation factor (equilibrium) model was customized for the Baltic Sea, as described by Bezhenar et al. (2016) and Maderich et al. (2018a). A set of nested boxes inside the regional box no. 68 in the Baltic Sea box system was added to resolve the near field of radionuclide concentration (Fig. 5). Parameters of the "inner box" and "coastal box" are based on data from (SKB, 2010). The main parameters of boxes with zooming in to the NPP are presented in Table S3





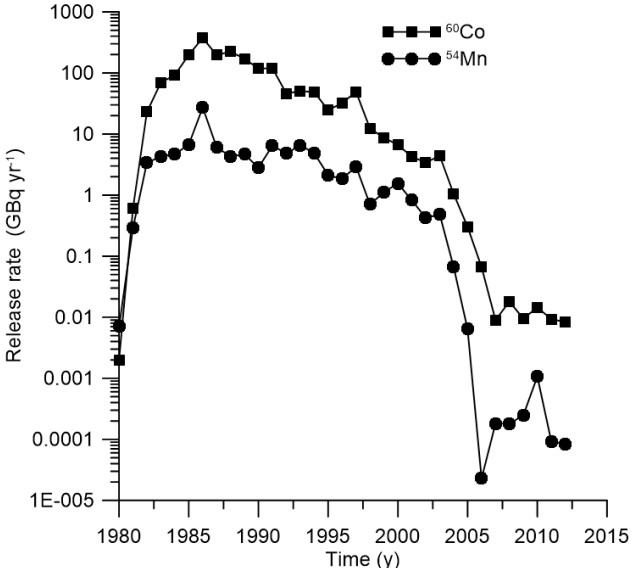

**Figure 6.** Release rates of $^{60}$Co and $^{54}$Mn according to measurements (Forsmark, 2014).

in the Supplement. The simulation results for $^{60}$Co and $^{54}$Mn were compared with measurements for the smallest coastal box,

where measurement data for bottom sediments and fish were available (Forsmark, 2014). The measurement data were compared with simulations for two species of fish: herring (*Clupea harengus membras*) as a non-piscivorous fish and pike (*Esox lucius*) as a coastal predator. There is no information on the mass of fish caught in the vicinity of Forsmark NPP. Therefore, we used estimates of the masses typical for prey and predatory fish, which are given in Table S1. In the one-compartment model, two parameters must be prescribed: assimilation efficiency and biological half-life $T_{0.5} = ln2\lambda_{wb}^{-1}$. Assimilation efficiency (see

Table 2) was obtained from Pouil et al. (2018). Baudin et al. (1997) used $T_{0.5} = 21$ d for $^{60}$Co in the one-compartment model. The average value for $T_{0.5}$ in predatory marine fish is 40 d (Beresford et al., 2015). Therefore, for $^{60}$Co, we used $T_{0.5} = 20$ d for prey fish and $T_{0.5} = 40$ d for predatory fish. There are very limited data for $T_{0.5}$ values in marine fish for $^{54}$Mn. According to Beresford et al. (2015), $T_{0.5}$ is in the range between 20 and 40 days. Therefore, we used the same values of $T_{0.5}$ for $^{54}$Mn, as for $^{60}$Co.

The release rates of $^{60}$Co and $^{54}$Mn from the Forsmark NPP (Forsmark, 2014) are plotted in Fig. 6. As seen in Fig. 7a, the results of simulation for the concentration of $^{60}$Co in the bottom sediments are in good agreement with the measurements (Forsmark, 2014) in the coastal box for the wide range of employed values of sediment distribution coefficient $K_d$: from $K_d = 3 \cdot 10^5 \div 2 \cdot 10^6$ L kg$^{-1}$ for margin seas to $K_d = 5 \cdot 10^7$ L kg$^{-1}$ for open ocean (IAEA, 2004). The benthic food web (Bezhenar et al., 2016a), which describes the transfer of radioactivity from bottom sediments to deposit-feeding invertebrates

and finally to fish, is quite important in this range of $K_d$ values. The results from the POSEIDON-R calculations obtained with the MCKA model are compared (Fig. 7b,c) with measurements and results of calculations obtained with a one-compartment model and with an equilibrium approach using a standard BAF value (IAEA, 2004). Whereas one-compartment and equilibrium





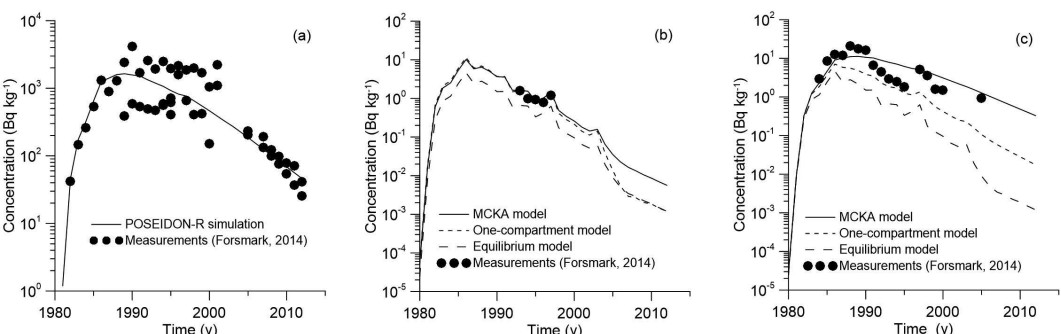

**Figure 7.** Comparison between calculated and measured $^{60}$Co (Forsmark, 2014) concentrations in bottom sediment (a), non-piscivorous fish (herring) (b) and coastal predator fish (pike) for the coastal box.

models underestimated the concentration of $^{60}$Co in fish, the MCKA model using generic parameters yields better agreement with measurements for both non-piscivorous fish (Fig. 7b) and coastal predator feeding by pelagic and benthic organisms in the coastal area (Fig. 7c).

Similarly, the behaviour of $^{54}$Mn in the marine environment near the Forsmark NPP is modelled. The release rate of $^{54}$Mn from the NPP is also plotted in Fig. 6 using data from Forsmark (2014). Comparison of the simulated concentration of $^{54}$Mn in bottom sediments with measurements (Forsmark, 2014) for the Forsmark coastal box is given in Fig. 8a. Good agreement was obtained when a standard value of $K_d = 2 \cdot 10^6$ L kg$^{-1}$ for $^{54}$Mn in margin seas (IAEA, 2004) was used. This means that, as in the case of $^{60}$Co, a significant fraction of radionuclide is deposited at the bottom, and the benthic food web should be considered. Similarly to the $^{60}$Co case, obtained results of simulation are also compared with results obtained using the one-compartment model and equilibrium approach. Again, the MCKA model yields the best agreement with measurements for both non-piscivorous fish (Fig. 8b) and coastal predators (Fig. 8c); however, this agreement is slightly worse than in the $^{60}$Co case. Notice that the BAF in the equilibrium approach can be locally estimated using *a posteriori* data. However, the MCKA model provided good agreement with measurements using only *a priori* information, which is important in the case of accidents, as considered in the next section.

## 4.3 Accumulation of $^{90}$Sr in the fish after the Fukushima Dai-ichi accident

Following caesium, $^{90}$Sr is the second most important radiologically long-lived radionuclide released as a result of the FDNPP accident. It is highly soluble in water and exhibits relatively high ability for assimilation by marine organisms due to similar chemical properties with calcium. The atmospheric deposition of $^{90}$Sr is usually not taken into account due to its low volatility. Most of the $^{90}$Sr released from the FDNPP was directly released to the ocean, with estimates of total inventory in the range from 0.04 to 1.0 PBq (Buesseler et al., 2017). Here, we extended the simulation by Maderich et al. (2014b) of transfer and fate of $^{90}$Sr resulting from the FDNPP accident using the POSEIDON-R model complemented by the food web model (Bezhenar et al., 2016) and MCKA fish model.





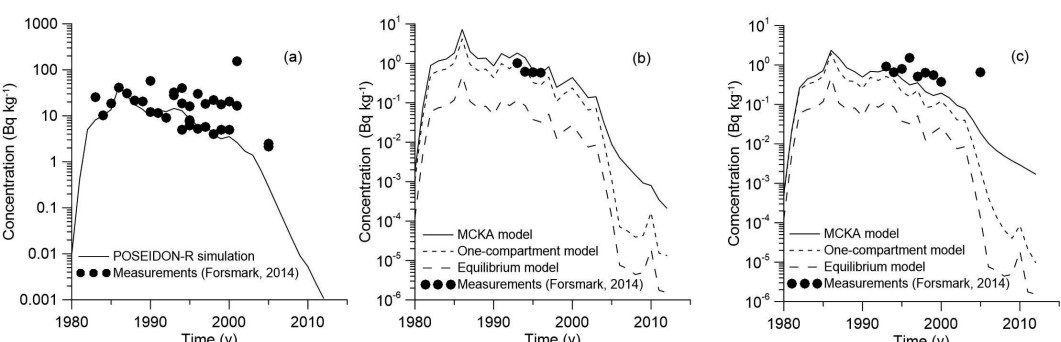

**Figure 8.** Comparison between calculated and measured $^{54}$Mn (Forsmark, 2014) concentrations in bottom sediment (a), non-piscivorous fish (herring) (b) and coastal predator fish (pike) for the coastal box.

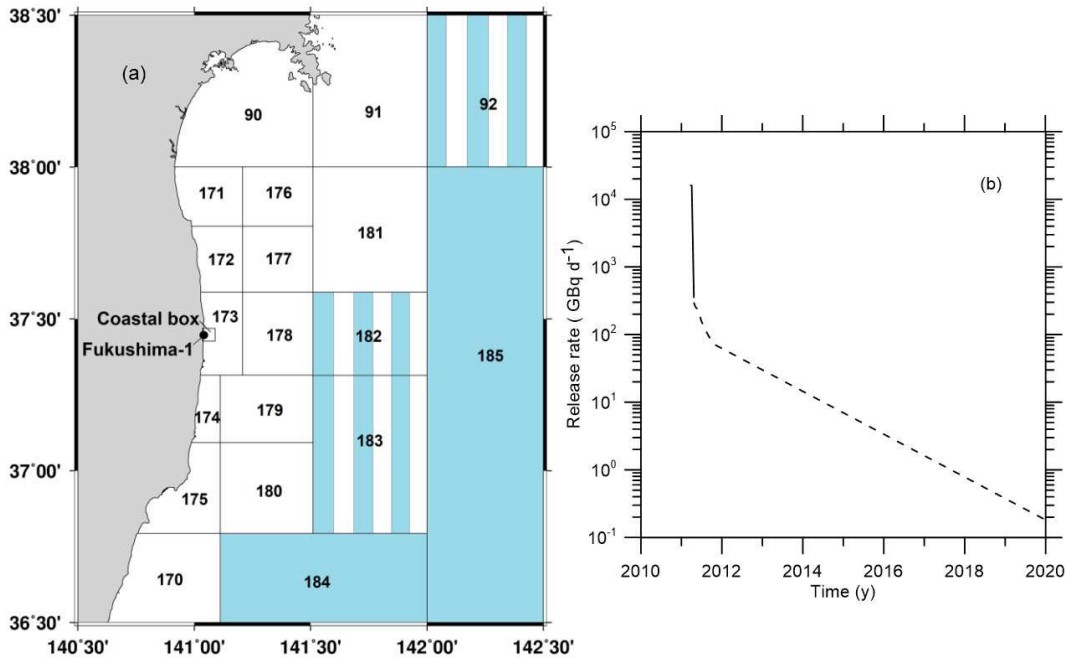

**Figure 9.** (a) Boxes along the eastern coast of Japan with fine resolution in the area around the FDNPP (Maderich et al., 2018a). Deep boxes with 3 vertical layers in the water column are coloured by blue, boxes with 2 vertical layers are marked by vertical stripes, and shallow one-layer boxes are white. (b) Release rates of $^{90}$Sr in the accidental (bold line) and post-accidental (dashed line) periods.

The POSEIDON-R model was customized for the Northwestern Pacific and adjacent seas (the East China Sea, the Yellow Sea and the East/Japan Sea) as in (Maderich et al., 2018a). A total of 188 compartments covered this region. The compartments around the FDNPP are shown on Fig. 9 with an additional 4x4 km coastal box in the vicinity of FDNPP. The coastal box was



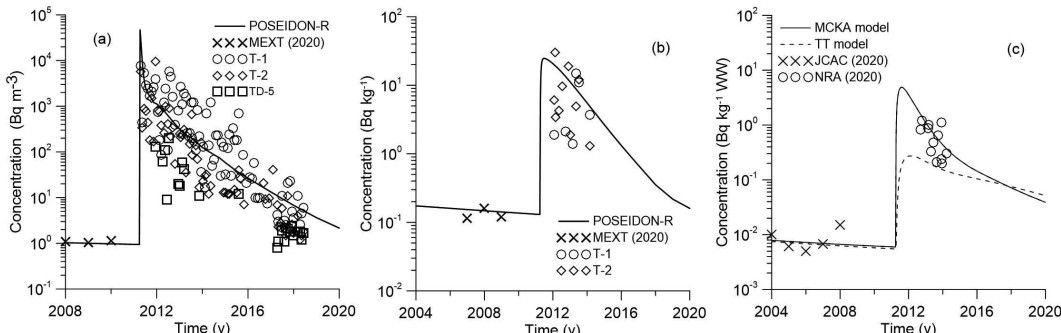

**Figure 10.** Comparison between calculated and measured $^{90}$Sr concentrations in water (a), bottom sediment (b), and piscivorous fish (c) for the coastal box.

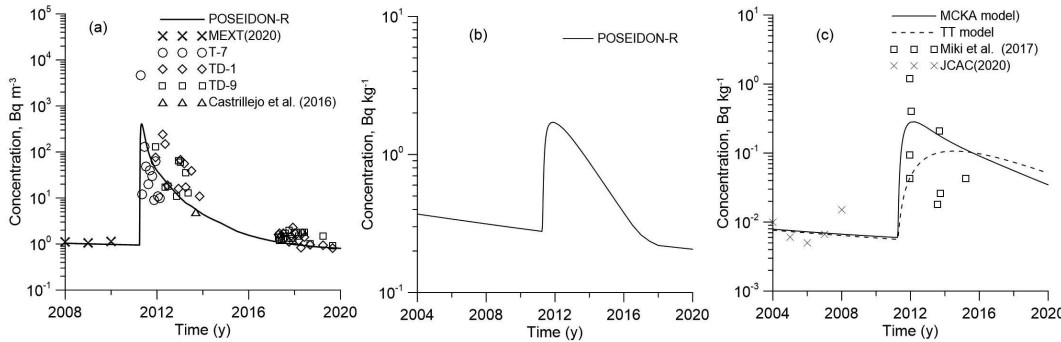

**Figure 11.** Comparison between calculated and measured $^{90}$Sr concentrations in water (a), bottom sediment (b), and piscivorous fish (c) for box no. 173.

included for the model validation with $^{90}$Sr measurements conducted in this specific area. Details of customization are given in Maderich et al. (2014a,b; 2018a).

325 The historical contamination due to global atmospheric deposition in the period from 1945-2010 was simulated according to (Maderich et al., 2014b) with data from the MARiS database (2020). The value of the accidental release was estimated as 160 TBq (16 TBq day$^{-1}$ during 10 days), that was consistent with the range reported by Buesseler et al. (2017). In the post-accidental period, the continuous leakage of $^{90}$Sr due to groundwater transport of radioactivity from the NPP site was monitored (Castrillejo et al., 2016). Therefore, in the simulation the conservative scenario was used for release of $^{90}$Sr in the

330 post-accidental period (Fig. 9); the release of $^{90}$Sr was assumed equal to $^{137}$Cs release (Maderich et al. (2018a). Comparison between calculated and measured $^{90}$Sr concentrations in water, bottom sediment, and piscivorous fish for the coastal box and box no. 173 are shown in Figs. 10 and 11, respectively. Measured concentrations of $^{90}$Sr in the water and bottom sediments before the accident were obtained from the MEXT database (MEXT, 2020). Concentrations of $^{90}$Sr after the accident at TEPCO (Tokyo Electric Power Company) sampling points near the discharging canals (T1 and T2) and at different distances offshore

335 (TD-5 inside the coastal box area, T7, TD-1 and TD-9 for outer box no. 173) are available in the NRA (Nuclear Regulation





Authority) database (NRA, 2020). The results of simulation show that the concentration of $^{90}$Sr in the seawater reaches the maximum just after the accidental release (Figs. 10a, 11a). Notice that the box model gives the average concentrations for each box, which means that local concentrations may be larger or smaller than the average (Fig. 10). Especially large differences can occur during the accidental release under strongly non-equilibrium conditions. Further large dispersion of measured con-

centrations shows that non-equilibrium conditions remained for a long time in the area close to NPP (Fig. 10a). The agreement between calculated and measured concentrations (Figs. 10a, 11a) could be a confirmation of the correctness of estimation of the source term. This is also confirmed by agreement of measured and simulated concentrations of $^{90}$Sr in the bottom sediment (Fig. 10b). Notice that there is no measurement data for bottom sediment in box no. 173 (Fig. 11b).

The calculated concentration of $^{90}$Sr in piscivorous fish was compared with measurement data for fat greenling (*Hexagram-*

*mos otakii*) before the accident (JCAC, 2020) and with data from (NRA, 2020) and Miki at al. (2017) after the accident. The NRA data for $^{90}$Sr are very limited. Therefore, different species of piscivorous fish were considered, such as sharks (*Triakis scyllium*, *Squatina japonica*), rockfish (*Sebastes cheni*) and seabass (*Lateolabrax japonicus*). The simulation results with the MCKA model agree well with the experimental observations (Fig. 10c,11c), while the target tissue (TT) approach underestimates the concentration of $^{90}$Sr in the fish.

**5    Conclusions**

A new approach to predicting the accumulation of radionuclides in fish taking into account heterogeneity of distribution of contamination in the organism and dependence of metabolic process rates on the fish mass was developed. The fish organism was represented by compartments for three groups of tissues/organs (muscle, bone, organs) and two input compartments representing gills and digestive tract. The absorbed elements are redistributed between organs/tissues and then eliminated according

to their metabolic function. The food and water uptake rates, elimination rate and growth rate depend on the metabolic rate, which is scaled by fish mass to the $3/4$ power. This model is of intermediate complexity and provides an alternative for the basic/simplistic whole-body models and the highly advanced PBPK models.

The trophic transfer factors ($TTF$) were calculated for 9 elements using assimilation efficiencies $AE_f$ obtained from laboratory data. Among considered elements, only caesium (Cs) may biologically magnify when transferring upwards into the

food chain ($TTF > 1$). This is in agreement with measurements. The concentrations of other elements (Sr, Co, Mn, Zn, Ag, Cu, Cd, and Cr) decrease with the increase of trophic level ($TTF < 1$). The kinetics of the assimilation and elimination of $^{134}$Cs, $^{57}$Co, $^{60}$Co, $^{54}$Mn and $^{65}$Zn, which are preferably accumulated in different tissues, were analyzed using the analytical solutions of a system of model equations. These solutions exhibited good agreement with the laboratory experiments for the depuration process after single feeding of fish with radiolabelled prey and with respect to uptake of activity from water. Notice

that for relatively slow processes in the marine environment, transfer processes in the gills and digestive tract can be close to equilibrium, which allows consideration of only a three-compartment (muscle, bone, organs) version of the model.

The developed MCKA model was embedded into the compartment model POSEIDON-R, which describes the transfer of radionuclides through the pelagic and benthic food webs. The POSEIDON-R model was applied for the simulation of the





transport and fate of $^{60}$Co and $^{54}$Mn routinely released from Forsmark NPP located on the Baltic Sea coast of Sweden and for

calculation of $^{90}$Sr concentration in fish after the accident at Fukushima Dai-ichi NPP. Predicted concentrations of radionuclides in fish agreed well with the measurements in both case studies. It is shown that the MCKA model with the defined generic parameters could be used in different marine environments without calibration based on *a posteriori* information, which is important for emergency decision support systems (Periáñez et al., 2019).

*Author contributions.*  RB, VM and KJ conducted the literature review and designed the study. RB, VM and KJ developed the model and

performed modelling of laboratory experiments. GdV and KK collected data for case studies. RB, VM and KK performed the simulations for case studies. RB, VM, KJ and GdV analysed results of simulations and wrote the initial manuscript. All authors edited and approved the final manuscript text

*Competing interests.*  The authors declare that they have no conflict of interest.

*Acknowledgements.*  This work was partially supported by KIOST major project PE99812, National Research Foundation of Ukraine projects

nos. 2020.02/0048 and 2020.01/0421, and IAEA Coordinated Research Project K41017.



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
