# Peer review of "Multi-compartment kinetic-allometric model of radionuclide bioaccumulation in marine fish"

_Biogeosciences, 2020_

## Referee Comment (RC1) · Anonymous Referee #1 · 17 Dec 2020

This paper reported the idea of radionuclide kinetic transfer model of tissue compartments (muscle, bone and organ) associated with growth of fish (multi-compartment kinetic allometric model: MCKA). The result of modelling tests demonstrated that the simulated temporal changes of 134Cs, 57Co, 54Mn and 65Zn levels in whole bodies and muscle of juvenile/adult sea bream, turbot and spotted dog fish reconstructed well the experimental results by Mathews and Fisher, 2008 and Mathews et al., 2008. The test result also exhibited that the bioconcentration factor (BCF) derived by simulation for 134Cs, 57, 60Co, 54Mn and 65Zn levels in whole bodies of juvenile sea bream and turbot agreed to the experimental results by Mathews and Fisher, 2008 and Jeffree et al., 2006. The applied results by MCKA model for temporal levels changes in fish of 60Co

and 54Mn at the vicinity of the Forsmark nuclear power plant of Baltic Sea, and 90Sr at Fukushima coasts were shown as being comparatively close to the measured whole-body concentrations in predator fishes than those generated from one-compartment model and tissue target model. The paper demonstrated that the MCKA model applicability to calculate the temporal changes of radionuclide levels in whole body of fish during 20 years. The approach method for evaluation of radionuclides levels in whole body was valuable to assessment of seafood safety in case of whole fish consumption, and possibly the radiation dose to wild life in the environment. The presented result may be worth to publish. However, the values of key parameters were not shown in the paper, which made reader being difficult to understand the rational sequence of modelling procedure. Especially of those bio-chemically different parameters for Cs, Sr and Co, Mn, Zn were not shown. It was insufficient only demonstrating the assimilation efficiency and the allometric parameters in the results. Because of these, the modeling methodology was not easy to understand and also the paper contents being vague. Therefore, the following four points are strongly recommended to revise before publish, to make the paper as being scientifically correct, and also helping reader's understanding.

1) Line 70: To help the reader's understanding, the resulted specific parameter values of ïĄňïĄľïĄ¡1-5. Kw, Kf, k1i=3-5, k2i=3-5 for Cs, Co, Mn, Zn, Sr has to be shown in supplementary Table. The parameter values of ïĄňg for sea bream, talbot, spotted dog fish, herring, pike also have to be shown in supplementary Table if they were decided as similar to AEw and AEf referred in line 214. 2) Line 115: The referred MCKA parameter values in Table S1 has to be associated with Cs, Co, Mn, Zn and Sr, because each metabolism was different resulting specific values. 3) Line 115: if Table S1 values only derived by mass difference of fish size, it has to be mentioned that "We did not consider the change of prey preference along growth in this study", which was referred in line 163-165. 4) Fig. 7 and 8: The salinity of area studied was 3-5 PSU, suggesting the estuary being close to freshwater environment. The description about how the author parameterize to simulate 60Co and 54Man level reconstruction marine fish herring and

freshwater fish pike under such low salinity brackish water environment.

Minor comments

Line 15: "Predicted" read as "Reconstructed" or "Computed".

Lune 16: "predicted" read as "calculated" or "computed".

Line 27: "effective recession times" read as "effective half-life".

Line 29: "Tateda et al., 2013" has to be delated from citation, because of the model is for target tissue (muscle).

Line 35: "Tateda et al., 2013" has to be added in citation, because of the model is for target tissue (muscle).

Line 38, Fig. 1: There were no data of body tissue mass in the referred Yankovitch et al., 2010 (no kidney CR data and body tissue ratio data). The exact citation has to be shown, or the calculation process for Fig. 1 has to be shown in the paper as supporting material.

Table 2: The values for Ag, Cu, Cd and Cr may be not necessary in this paper because of this paper result only demonstrated the simulations of Cs, Co, Mn, An and Sr.

Line 163-165: The description of "The BAF . . .our findings" has to be re-considered, because the modelling in this paper seems not include the change of prey-type associated with fish growth.

Line 165: "1999" read as "1995"

Line 189: "however, . . .greater in the muscle" has to be reconsidered, because the retained levels of blue line A3/Af (muscle) were higher than A4/Af (bone) and A5/Af (organs) for all four nuclides in Fig. 2.

Fig. 4: "Co" read as "57Co and 60Co".

Fig. 4: The model simulated results of dog fish were not shown.

Line 425: "1999" reads as "1995"

Line 359: "may biologically magnify when transferring upwards into the food chain" read as "level may elevate in the predator fish of the food chain", because Cs was not accumulative element compared to Hg and Cd.

*** The presented modeling methodology and logic to draw conclusion are reasonable and worth to be published for the Journal paper. However, the defined parameter were not shown in the paper which makes the paper content being not transparent and verifiable. Therefore, the manuscript has to be revised by the listed comments before accept as the paper for this journal.

––––––––––––––––––––––––––

---

## Author Comment (AC1) · 19 Jan 2021

Dear Reviewer, the authors are most grateful to you for thorough analysis of manuscript and for constructive criticism and suggestions. We have taken all yours remarks into account, and the paper has been revised in many places accordingly. Thank you, Vladimir Maderich Please also note the supplement bg-2020-423-supplement.pdf to this comment:

Please also note the supplement to this comment:
https://bg.copernicus.org/preprints/bg-2020-423/bg-2020-423-AC1-supplement.pdf

[Figure]

[Figure]

**Supplement:**

**Response to Reviewer #1**

The authors are most grateful to the reviewer for thorough analysis of manuscript and for constructive criticism and suggestions. We have taken his remarks into account, and the paper has been revised in many places accordingly.

*General comments*

*This paper reported the idea of radionuclide kinetic transfer model of tissue compartments (muscle, bone and organ) associated with growth of fish (multi-compartment kinetic allometric model: MCKA). The result of modelling tests demonstrated that the simulated temporal changes of $^{134}$Cs, $^{57}$Co, $^{54}$Mn and $^{65}$Zn levels in whole bodies and muscle of juvenile/adult sea bream, turbot and spotted dog fish reconstructed well the experimental results by Mathews and Fisher, 2008 and Mathews et al., 2008. The test result also exhibited that the bioconcentration factor (BCF) derived by simulation for $^{134}$Cs, $^{57, 60}$Co, $^{54}$Mn and $^{65}$Zn levels in whole bodies of juvenile sea bream and turbot agreed to the experimental results by Mathews and Fisher, 2008 and Jeffree et al., 2006. The applied results by MCKA model for temporal levels changes in fish of $^{60}$Co and $^{54}$Mn at the vicinity of the Forsmark nuclear power plant of Baltic Sea, and $^{90}$Sr at Fukushima coasts were shown as being comparatively close to the measured wholebody concentrations in predator fishes than those generated from one-compartment model and tissue target model. The paper demonstrated that the MCKA model applicability to calculate the temporal changes of radionuclide levels in whole body of fish during 20 years. The approach method for evaluation of radionuclides levels in whole body was valuable to assessment of seafood safety in case of whole fish consumption, and possibly the radiation dose to wild life in the environment. The presented result may be worth to publish. However, the values of key parameters were not shown in the paper, which made reader being difficult to understand the rational sequence of modelling procedure. Especially of those bio-chemically different parameters for Cs, Sr and Co, Mn, Zn were not shown. It was insufficient only demonstrating the assimilation efficiency and the allometric parameters in the results. Because of these, the modeling methodology was not easy to understand and also the paper contents being vague. Therefore, the following four points are strongly recommended to revise before publish, to make the paper as being scientifically correct, and also helping reader's understanding.*

**Answer.** We modified accordingly Table 2 including data for $AE_w$. The tables with MCKA model parameters and transfer rates for each laboratory experiment and for case studies were

added in the Supplemenary Material (Tables S1-S8). The content of these tables is discussed below.

*1) Line 70: To help the reader's understanding, the resulted specific parameter values of ïA¸nˇ ïA¸l'ïAˇ¡1-5. $K_w$, $K_f$, $k_{1i}$=3-5, $k_{2i}$=3-5 for Cs, Co, Mn, Zn, Sr has to be shown in supplementary Table. The parameter values of ïA¸nˇg for sea bream, talbot, spotted dog fish, herring, pike also have to be shown in supplementary Table if they were decided as similar to $AE_w$ and $AE_f$ referred in line 214.*

**Answer**. We added tables with MCKA model parameters (Tables S1 and S5) and transfer rates $k_{2,i}$ and $k_{1,i}$ (Tables S2-S4 and S6-S8) for each fish in laboratory experiments and Table S11 with parameters of MCKA model for prey fish and predator fish in the marine case studies. The text was changed accordingly:

Line 201 "Parameters of MCKA model for fish from experiments (Mathews and Fisher, 2008; Mathews et al., 2008) are given in Table S1, whereas Tables S2-S4 show dependence on radionuclides of the transfer rates $k_{2,i}$ in different fishes."

Line 215 "Parameters of MCKA model for these fishes are given in Table S5, whereas Tables S6-S8 show dependence on radionuclides of the transfer rates $k_{2,i}$ in different fishes."

Line 296 "Therefore, we can apply the model parameters defined for marine environment (see Table S11) to reconstruct the herring and pike contamination by the above-mentioned radionuclides in the area near the Forsmark NPP with low salinity (3-5 PSU)."

Line 356 "Parameters of MCKA model for these fishes are given in Table S11."

*2) Line 115: The referred MCKA parameter values in Table S1 has to be associated with Cs, Co, Mn, Zn and Sr, because each metabolism was different resulting specific values.*

**Answer.** Parameter values in Tables S1, S5 and S11 (food uptake rate $K_f$, water uptake rate $K_w$ and elimination rates $\lambda_i$) depend on the mass of fish and do not depend on radionuclide, whereas an activity is distributed between different tissues/organs according to assimilation efficiencies (Table 2), which are different for different radionuclides. The combination of all these parameters defines the processes of radionuclides uptake, retention and elimination that leads to the differences of fish contamination by each radionuclide. The text was added accordingly:

Line 364 "The food and water uptake rates, elimination rate and growth rate depend on the metabolic rate, which is scaled by fish mass to the 3/4 power, but do not depend on the radionuclide. At the same time, the activity is distributed between different tissues/organs according to the tissue assimilation efficiencies, which are different for different radionuclides

(Table 2), but does not dependent on fish mass. Therefore, the transfer rates can be associated with specific radionuclide and fish mass as shown e.g. in Tables S2-S4."

*3) Line 115: if Table S1 values only derived by mass difference of fish size, it has to be mentioned that "We did not consider the change of prey preference along growth in this study", which was referred in line 163-165.*

**Answer.** The text was changed accordingly:

Line 165 "The BAF in larger and older fish of the same species can differ from smaller and younger fish due to the change of habitat and diet with age (e.g. Kasamatsu and Ishikawa, 1997; Ishikawa et al., 1995; Kim et al., 2019), however, in this study we did not consider the change of prey preference along the fish growth."

*4) Fig. 7 and 8: The salinity of area studied was 3-5 PSU, suggesting the estuary being close to freshwater environment. The description about how the author parameterize to simulate 60Co and 54Man level reconstruction marine fish herring and freshwater fish pike under such low salinity brackish water environment.*

**Answer.** We added text accordingly:

Line 295 "According to Jeffree et al. (2017), the uptake and depuration kinetics of $^{60}$Co and $^{54}$Mn for fish species in marine, brackish and freshwater environments are similar.

Therefore, we can apply the model parameters defined for marine environment (see Table S11) to reconstruct the herring and pike contamination by the above-mentioned radionuclides in the area near the Forsmark NPP with low salinity (3-5 PSU)

***Minor comments***

*Line 15: "Predicted" read as "Reconstructed" or "Computed".*

**Answer.** Done

*Lune 16: "predicted" read as "calculated" or "computed".*

**Answer.** Done

*Line 27: "effective recession times" read as "effective half-life".*

**Answer.** Done

*Line 29: "Tateda et al., 2013" has to be deleted from citation, because of the model is for target tissue (muscle).*

**Answer.** Done

*Line 35: "Tateda et al., 2013" has to be added in citation, because of the model is for target tissue (muscle).*

**Answer.** Done

*Line 38, Fig. 1: There were no data of body tissue mass in the referred Yankovitch et al., 2010 (no kidney CR data and body tissue ratio data). The exact citation has to be shown, or the calculation process for Fig. 1 has to be shown in the paper as supporting material.*

**Answer.** Data for kidney were removed from the Fig. 1. We added reference on Yankovich (2003) where detailed data on tissue mass fractions are reported. The percentage of activity in a given tissue $F_i$ was calculated as a ratio of tissue mass fraction to whole body $\mu_i$ to tissue concentration ratio $CR_i$ multiplied by 100%: $F_i = \mu_i / CR_i \cdot 100\%$. Corresponding changes were made on the Fig. 1 and in the text.

Line 38 "Distribution of accumulated activities of isotopes Cs, Sr and Co in muscle, bone and liver estimated from previously reported data (Yankovich, 2003; Yankovich et al., 2010) are shown in Fig. 1. The accumulated activity in a given tissue was calculated as a ratio of tissue mass fraction (%) (Yankovich, 2003) to body-to-tissue concentration ratio (Yankovich et al., 2010).

*Table 2: The values for Ag, Cu, Cd and Cr may be not necessary in this paper because of this paper result only demonstrated the simulations of Cs, Co, Mn, An and Sr.*

**Answer.** Done.

*Line 163-165: The description of "The BAF : : :our findings" has to be re-considered, because the modelling in this paper seems not include the change of prey-type associated with fish growth.*

**Answer.** See answer on General comment #3.

*Line 165: "1999" read as "1995"*

**Answer.** Done

*Line 189: "however, : : :greater in the muscle" has to be reconsidered, because the retained levels of blue line A3/Af (muscle) were higher than A4/Af (bone) and A5/Af (organs) for all four nuclides in Fig. 2.*

**Answer.** The curves in Fig. 2 represent activity in the tissues normalized on the total amount of ingested activity. Most of the activity is contained in muscle (blue line in Fig. 2). However, first 5 days after feeding the concentrations of [60]Co and [54]Mn in organs are much greater than concentrations in the muscle. We have adjusted the description of the figure accordingly:

Line 190 "However, the first 5 days after feeding the concentrations of [60]Co and [54]Mn in the organs are much greater than in the muscle."

*Fig. 4: "Co" read as "57Co and 60Co".*

**Answer.** Done

*Fig. 4: The model simulated results of dog fish were not shown.*

**Answer.** Parameters of MCKA model for *Psetta maxima* and *Scyliorhinus canicula* from experiments for uptake of activity from sea water (Table S5) are very close, therefore computed curves for *Psetta maxima* and *Scyliorhinus canicula* in Fig. 4 almost coincide.

*Line 425: "1999" reads as "1995"*

**Answer.** Done

*Line 359: "may biologically magnify when transferring upwards into the food chain" read as "level may elevate in the predator fish of the food chain", because Cs was not accumulative element compared to Hg and Cd.*

**Answer.** Done.

---

## Referee Comment (RC2) · Anonymous Referee #2 · 23 Feb 2021

The authors present a multi-compartment model for radionuclide bioaccumulation in fish. The compartments for this model are muscle, bones, and organs. Uptake can be by direct absorption through gills, or from food. Transfer is also allowed between compartments. The model was tested on a set of radionuclides, with good agreement with lab experiments. The model was implemented into the POSEIDON-R, and applied to several real-world scenarios. This seems to work better than the previous single-compartment model that was previously used.

Overall, the paper is detailed and well-written. The authors present a novel method, integrated into a current software with applications to real-world problems.

[Figure]

My major concern about the work here is in the comparison between the MCKA model and one-compartment or equilibrium models. The Forsmark results seem to show improvements in estimates from the MCKA model as opposed to the one-compartment model and Equilibrium models. However, I'm not convinced that it's not just because of poor-quality estimates of parameters for the one-compartment model and equilibrium. The equilibrium model consistently underpredicts by a factor of $\sim$10 for 54Mn over a period of decades. If you want a fair comparison for the underlying model, then you need to make sure they all the parameters are consistent. Are the parameters consistent between models? That is, you could use the MCKA model to estimate equivalent one-compartment parameters and BAF parameters such that the equilibrium concentrations are all identical. In that case, are the results significantly different? If the results are still different, then you have shown that your additional model complexity is needed for higher accuracy in these dynamic problems. If they aren't, then it just shows that your method can be used to estimate these factors for a given ecosystem model. This would still be an excellent finding, as it will help with model building, but it wouldn't be necessary to explicitly track all the concentrations inside the model. Judging from the results in figure 2, it looks like the inter-compartment equilibrium is reached quite quickly (<2 days?) in this case, the system should behave identically to a single-compartment system, should it not?

If this issue is resolved then I would highly recommend publication.

Specific comments:

l. 127 Not sure what is meant by "The equations under (17) is used"

Should define BAF in equation somewhere. You may also note that IAEA uses concentration ratio CR or concentration factor CF to describe what you are using as bioaccumulation factor BAF, while you use CR for something different.

I believe the IAEA document only has BAF_wb, not BAF_food, does it not?

l. 229 "shown the importance of including in the model of the digestive tract compartment describing highly non-equilibrium transfer dynamics" this seems to show the importance of kinetics in the modeling, but not the digestive compartment per se, as opposed to just using a single compartment.

The half-life of 54Mn is only 312 days, so could be relevant compared to the biological half-lives. Was this accounted for in the modeling>?

l. 321 using compartments here as spatial regions may be confusing.

Technical corrections:

Figure 1 should be regenerated in higher-quality.

Regarding eq. 1-3, you describe all variables except $C_i$

l. 125 should be lambda_g

Figures 2-4 are low quality JPG. Avoid using lossy compression (jpg) on graphs – use lossless (e.g., png) or vector graphics (pdf/svg/wmf).

l. 240 space before 60Co

---

## Referee Comment (RC3) · Anonymous Referee #3 · 3 Mar 2021

The authors present a multi-compartment kinetic-allometric model for radionuclide bioaccumulation in fish. First, the authors present the development of the model. They tested their model on data from laboratory experiments and several radionuclides. Finally, they used their model to simulate real-case scenarios. I believe that the development of such model could have strong contribution for the risk assessment of radionuclides. However, the manuscript would need some clarifications before considering it for publication.

General comments:

The authors should give more information on the MCKA model structure: I would sug-

gest adding a schematic representation of the model (In part 2. Model) as it would be of a great help to visualize the structure of the model and better understand the relationships between the compartments. The authors should also better present the parameters of the model by given a table with the definition of all parameters, their values and units. Especially, the values parameter related to the radionuclides should be clearly presented. Table 1 only presents the parameters in allometric relations.

The last part (Part 4. Model applications) is really interesting as it presents model applications for simulating radionuclide bioaccumulation in real contexts. This part may even represent another manuscript. I understand why the authors wanted to keep this part here, but more explanations should be added. The relationships between the fish and their prey should be better explained. How do you handle the preference type (Pj) in the MCKA models? Could you give the parameter values? Also, you discussed that there was no major impact of temperature on uptake and elimination in laboratory experiments, but what about in real-case scenarios (l. 110)? Similarly, what about salinity? Are the abiotic parameters taken into account in the POSEIDON-R model? Without more information, the coupling between the POSEIDON-R model and the MCKA model is still unclear, and the simulation results could be questioned.

They also compare the results to a one-compartment model that was not presented in the simulation cases before, hence it is hard to conclude on the comparison with this model. Have you compared the simulations of the one-compartment model and your MCKA model for laboratory experiments? It would help to better explain the importance of adding complexity in the MCKA model compared to a single-compartment model.

With more clarifications on those different points, I believe the manuscript would be worth for publication.

Specific comments:

- In your introduction, you do not mention the POSEIDON-R model that you used to do the simulations of the accidental releases, you should at least present it. Without

reading the abstract, you do not expect to have a coupling of two models in the last part. More globally, I am not use to this type of presentation of an article, that is why I was a little bit confused reading the manuscript. I felt more like reading a report, even if I understand the importance of each part.

- l. 48: I am not a specialist of PBPK models but you wrote that fish PBPK models do not include scaling allometric relationship between metabolic rate and organism mass However, the PBPK model of Grech et al. 2019 which takes into account the effect of growth on the cardiac output and oxygen consumption rate.

- l. 107: Could it have not be possible to adapt the dynamic budget theory (DEB) to model this? Could you explain better why this value of $\frac{3}{4}$ power? It is specific to fish?

- l.115. So, if I understand well, the structure of the model if generic for different fish species but the parameters values are specific to the species depending on their weight? Maybe you should mention it for more clearly. For two different fish species of the same weight, could you not have an inter-species variability of the model parameters?

- Fig 4. Why is there no curve for dogfish? If you could not simulate BCF for dogfish, maybe you should withdraw the concerning data points.

- Mathews et al. 2008 (l. 190) should not be Mathews and Fisher 2008? (l. 208)

- l. 242. A graphical representation of the results of the sensitivity analysis in SI rather than a table would be of a great help to clarify the results. Why did not you make a global sensitivity analysis to better understand interactions between parameters?

- l. 275. A schematic representation of the one-compartment model would be required in SI as well as a table with the parameter values.

- l. 276. I do not understand the sentence.

- l. 277. What are the "inner box" and "coastal box" exactly? I do not really understand,

as I am not familiar with the POSEIDON-R model. It should be better clarified.

- Figures 7 and 8 could be coupled (6 panels).

Technical corrections:

- Several spaces are missing: l. 107, l. 241

- Error on the reference: l. 278

---

## Author Comment (AC2) · 4 Mar 2021

**Response to Reviewer #2**

The authors are most grateful to the reviewer for thorough analysis of the manuscript and the constructive criticism and suggestions. We have followed your suggestions and revised the manuscript accordingly. Please, find our responses below.

*General comments*
*The authors present a multi-compartment model for radionuclide bioaccumulation in fish. The compartments for this model are muscle, bones, and organs. Uptake can be by direct absorption through gills, or from food. Transfer is also allowed between compartments. The model was tested on a set of radionuclides, with good agreement with lab experiments. The model was implemented into the POSEIDON-R, and applied to several real-world scenarios. This seems to work better than the previous single compartment model that was previously used. Overall, the paper is detailed and well-written. The authors present a novel method, integrated into a current software with applications to real-world problems.*

*My major concern about the work here is in the comparison between the MCKA model and one-compartment or equilibrium models. The Forsmark results seem to show improvements in estimates from the MCKA model as opposed to the one-compartment model and Equilibrium models. However, I'm not convinced that it's not just because of poor-quality estimates of parameters for the one-compartment model and equilibrium. The equilibrium model consistently underpredicts by a factor of _10 for 54Mn over a period of decades. If you want a fair comparison for the underlying model, then you need to make sure they all the parameters are consistent. Are the parameters consistent between models? That is, you could use the MCKA model to estimate equivalent one-compartment parameters and BAF parameters such that the equilibrium concentrations are all identical. In that case, are the results significantly different? If the results are still different, then you have shown that your additional model complexity is needed for higher accuracy in these dynamic problems. If they aren't, then it just shows that your method can be used to estimate these factors for a given ecosystem model. This would still be an excellent finding, as it will help with model building, but it wouldn't be necessary to explicitly track all the concentrations inside the model. Judging from the results in figure 2, it looks like the inter-compartment equilibrium is reached quite quickly (<2 days?) in this case, the system should behave identically to a single-compartment system, should it not? If this issue is resolved then I would highly recommend publication.*

**Answer.** Thank you for the discussion and the important suggestions.
(i) For the Forsmark simulation we used generic parameter values for all models including the MCKA. The aim was to demonstrate the ability of using the MCKA model without *a posteriori* information.
(ii) As follows from a comparison of the equations (1)-(3) and (7) of the MCKA model and the equation (11) of the standard whole-body model, the main difference between them is the description of the whole-body elimination rate $\lambda_{wb}$. Therefore, we considered in more detail the timely variations of the calculated $\lambda_{wb}$ in the MCKA model. The results are presented in a new Fig. 11.
(iii) Following your suggestion we compared simulation results on the FDNPP accident obtained with the MCKA model and the one-compartment model. In both models identical AE values are used. The $\lambda_{wb}$ computed using the MCKA model was variable, while the $\lambda_{wb}$ in the one-compartment model was set to a fix  value of 0.0027 d$^{-1}$ for piscivorous fish. This value was

obtained from the MCKA model representing the conditions prior to 2011 (Figure 11 (c)). The results from this comparison were added in Figs. 10 and 11, which were merged as Fig. 10.
(iv) We fixed some errors in the calculations presented in Fig. 2. Therefore, input of the different tissues is clearly visible now.

The text and figures were reworked as follows:

l. 364 "Comparison of the equations (1)-(3) and (7) of the MCKA model and the equation (11) of the standard whole-body model demonstrates that the main difference is found in the description of the whole-body elimination rate $\lambda_{wb}$. Whereas in the whole-body model the value of $\lambda_{wb}$ is constant, in the MCKA model it is the ratio of activity weighted tissue elimination rates to whole-body activity

$$\lambda_{wb} = \frac{\sum_{i=3}^{5} \mu_i \lambda_i C_i}{C_{wb}}. \tag{34}$$

Therefore, in the MCKA model, the value of $\lambda_{wb}$ can vary over time, depending on the uptake of radionuclide and the tissue elimination rates. The time variation of $\lambda_{wb}$ computed from (34) is shown in Fig. 11 for three different cases: (a) pulse-like feeding experiment (Matthews, Fisher (2008), (b) release of $^{60}$Co during normal operation of the Forsmark NPP and (c) accumulation of $^{90}$Sr in the fish due to the FDNPP accident. As seen in the plots, $\lambda_{wb}$ varies considerably when there is non-equilibrium, such as in the case of a pulse-like feeding or an accident. Even in case of a routine release, $\lambda_{wb}$ follows any changes in the release rate (Fig. 11b). In case of the FDNPP accident, the calculated $\lambda_{wb}$ shows some tendency towards an equilibrium value, but after a pulse-like release of $^{90}$Sr in 2011 $\lambda_{wb}$ doubled following the release of activity and then slowly converged to the quasi-equilibrium state governed by the global deposition. Notice that in this case we extended simulation period to 2040 extrapolating deposition data and FDNPP release data in Fig.9b. Therefore, whole-body model with a constant $\lambda_{wb}$, that is calibrated using observational data, cannot to correctly describe such transient processes in the organism. This is confirmed in the Fig. 10 by comparing the results from the MCKA model and the one-compartment model. Here the AE value in both two models are the same, whereas the equilibrium value $\lambda_{wb}$ is calculated in the MCKA model using value before 2011 for piscivorous fish ($\lambda_{wb}$ =0.0027 d$^{-1}$). As seen in Figs. 10c and 10f, the target tissue (TT) model (Maderich et al., 2015) underestimates the concentration in fish comparatively with MCKA model and observations. The one-compartment model simulation results using parameters from MCKA model are close to the MCKA model results at initial stage of accidental release. However, over time, the concentration $^{90}$Sr in fish tends to equilibrium faster than the MCKA model predicts, which is explained by the time-dependent behavior of $\lambda_{wb}$ in the MCKA model. The calculation results confirm that generic parameters of the MCKA model make it possible to correctly estimate $\lambda_{wb}$ without preliminary calibration on the local measurement data, which may be impossible in an accident. "

l. 396 "The main difference between MCKA and whole-body models was found in the description of the whole-body elimination rate $\lambda_{wb}$. Whereas in the whole-body model the value of $\lambda_{wb}$ is constant, in the MCKA model it is the ratio of activity weighted tissue elimination rates to whole-body activity as described (34). The elimination rate $\lambda_{wb}$ varies considerably in non-equilibrium state of fish, such as in the case of a pulse-like feeding or an accident."

[Figure]

**Figure 10.** Comparison between calculated and measured $^{90}$Sr concentrations in water (a), bottom sediment (b), piscivorous fish (c) for the coastal box and in water (d), bottom sediment (e), and piscivorous fish (f) for box no. 173.

[Figure]

**Figure 11.** The calculated $\lambda_{wb}$ for three scenarios: (a) the pulse-like feeding experiment (Matthews, Fisher, 2008), (b) the release of $^{60}$Co during normal operation of the Forsmark NPP and (c) the accumulation of $^{90}$Sr in the fish (coastal box) due to the FDNPP accident.

[Figure]

**Figure 2.** Retention of radionuclides in whole body and tissues of juvenile sea bream (*Sparus auratus*). The simulations are compared with whole body measurements by Mathews and Fisher (2008).

***Specific comments:***

*l. 127 Not sure what is meant by "The equations under (17) is used"*
**Answer.** The text was changed as
l. 131 "The equations (17) are used"

*Should define BAF in equation somewhere. You may also note that IAEA uses concentration ratio CR or concentration factor CF to describe what you are using as bioaccumulation factor BAF, while you use CR for something different. I believe the IAEA document only has BAF_wb, not BAF_food, does it not?*
**Answer:** We changed text accordingly.
l.139 "We define bioconcentration factor (*BCF*) as ratio of whole-body of fish to water concentrations with no dietary intake, bioaccumulation factor (*BAF*) as ratio of whole-body of fish to water concentrations with dietary intake, body-to-tissue concentration ratio ($CR_i$) as ratio of whole body to tissue concentrations, whereas ratio of food to water concentrations is indicated as $BAF_{food}$."
l.139 "Values of *BAF* for different radionuclides expressed as *CF* in (IAEA, 2004)."

l. 229 "*shown the importance of including in the model of the digestive tract compartment describing highly non-equilibrium transfer dynamics*" this seems to show the importance of kinetics in the modeling, but not the digestive compartment per se, as opposed to just using a single compartment.
**Answer.** Thank you for suggestion. The text in l. 229 was rewritten accordingly:

l. 239 "Comparison of the model against laboratory experiments on the retention of absorbed elements in fish after single feeding demonstrated the need to include the kinetic characteristics of the digestive tract in the model when highly non-equilibrium transfer dynamics are expected."

*The half-life of 54Mn is only 312 days, so could be relevant compared to the biological half-lives. Was this accounted for in the modeling>?*
**Answer:** The simplified equilibrium relations (17), which allow use of scaling (12) in (18) were obtained with the assumption that the elimination rates $\lambda_i$ were much greater than the physical decay rates $\lambda$. Therefore, the physical decay of radionuclides was not taken into account in the transfer processes in fish. For $^{54}$Mn, the decay constant $\lambda=0.0022$ was still less than the average value for fish $\lambda_{wb}=0.009$ in the Forsmark case study. Notice, that the physical decay was taken into account for processes of radionuclide transfer in the water and interaction with suspended and bottom sediments within the POSEIDON-R model used for long term simulation in the Forsmark and in the FDNPP case studies.

The text was added accordingly:
l. 131"This assumption also imposes requirement on the modelling of radionuclides with decay constant $\lambda<<\lambda_i$."

l. 321 *using compartments here as spatial regions may be confusing.*
**Answer:** Spatial "compartments" in text were replaced by "boxes"

*Technical corrections:*

*Figure 1 should be regenerated in higher-quality.*
**Answer.** Done

*Regarding eq. 1-3, you describe all variables except C_i*
**Answer.** Done.

l. 125 *should be lambda_g*
**Answer.** Done.

*Figures 2-4 are low quality JPG. Avoid using lossy compression (jpg) on graphs – use lossless (e.g., png) or vector graphics (pdf/svg/wmf).*
**Answer.** We used eps for all figures. These figures were reworked in higher quality.

l. 240 *space before 60Co*
**Answer.** Done.

---

## Author Comment (AC3) · 8 Mar 2021

**Response to Reviewer #3**

The authors appreciate the reviewer's valuable comments. They are very helpful for improving our paper. We have now modified the manuscript accordingly, and the changes are noted point by point. Please, find our responses below.

*The authors present a multi-compartment kinetic-allometric model for radionuclide bioaccumulation in fish. First, the authors present the development of the model. They tested their model on data from laboratory experiments and several radionuclides. Finally, they used their model to simulate real-case scenarios. I believe that the development of such model could have strong contribution for the risk assessment of radionuclides. However, the manuscript would need some clarifications before considering it for publication.*

**General comments**
*The authors should give more information on the MCKA model structure: I would suggest adding a schematic representation of the model (In part 2. Model) as it would be of a great help to visualize the structure of the model and better understand the relationships between the compartments. The authors should also better present the parameters of the model by given a table with the definition of all parameters, their values and units. Especially, the values parameter related to the radionuclides should be clearly presented. Table 1 only presents the parameters in allometric relations. The last part (Part 4. Model applications) is really interesting as it presents model applications for simulating radionuclide bioaccumulation in real contexts. This part may even represent another manuscript. I understand why the authors wanted to keep this part here, but more explanations should be added. The relationships between the fish and their prey should be better explained. How do you handle the preference type (Pj) in the MCKA models? Could you give the parameter values? Also, you discussed that there was no major impact of temperature on uptake and elimination in laboratory experiments, but what about in real-case scenarios (l. 110)? Similarly, what about salinity? Are the abiotic parameters taken into account in the POSEIDONR model? Without more information, the coupling between the POSEIDON-R model and the MCKA model is still unclear, and the simulation results could be questioned. They also compare the results to a one-compartment model that was not presented in the simulation cases before, hence it is hard to conclude on the comparison with this model. Have you compared the simulations of the one-compartment model and your MCKA model for laboratory experiments? It would help to better explain the importance of adding complexity in the MCKA model compared to a single-compartment model. With more clarifications on those different points, I believe the manuscript would be worth for publication.*

**Answer.** Thank you for the discussion and the important suggestions.

(1) We added a schematic description of the model in Fig. 2 and included the following text:
l. 74 "A schematic representation of the model is shown in Fig. 2."

(2) Following a suggestion by Reviewer#1 we updated the tables with model parameters. The model parameters are presented in Tables 1 and 2, and in several tables in the Supplementary Material:

**Table S1** Parameters of the MCKA model for fish used to simulate the experiments (Mathews and Fisher, 2008; Mathews et al., 2008) on pulse-like feeding.
**Table S2**. Transfer rate $k_{2,3}$ (d$^{-1}$) for radionuclides used to simulate the experiments (Mathews and Fisher, 2008; Mathews et al., 2008) on pulse-like feeding.

**Table S3**. Transfer rate $k_{2,4}$ ($d^{-1}$) for radionuclides used to simulate the experiments (Mathews and Fisher, 2008; Mathews et al., 2008) on pulse-like feeding.
**Table S4**. Transfer rate $k_{2,5}$ ($d^{-1}$) for radionuclides used to simulate the experiments (Mathews and Fisher, 2008; Mathews et al., 2008) on pulse-like feeding.
**Table S5**. Parameters of MCKA model for fish used to simulate the experiments (Mathews and Fisher, 2008; Jeffree et al., 2006) on uptake of activity from sea water.
**Table S6** Transfer rate $k_{1,3}$ ($d^{-1}$) for radionuclides used to simulate the experiments (Mathews and Fisher, 2008; Jeffree et al., 2006) on uptake of activity from sea water.
**Table S7** Transfer rate $k_{1,4}$ ($d^{-1}$) for radionuclides used to simulate the experiments (Mathews and Fisher, 2008; Jeffree et al., 2006) on uptake of activity from sea water.
**Table S8** Transfer rate $k_{1,5}$ ($d^{-1}$) for radionuclides used to simulate the experiments (Mathews and Fisher, 2008; Jeffree et al., 2006) on uptake of activity from sea water.
**Table S9**. Food preference $P_{ij}$ for predator of type $i$, and prey of type $j$.
**Table S11** Parameters of the MCKA model for prey fish (*Clupea harengus membras*) and predator fish (*Esox lucius, Hexagrammos otakii, Triakis scyllium, Squatina japonica, Sebastes cheni, Lateolabrax japonicus*) in POSEIDON-R model applications.

(3) We added a table (Table S9) for food preference in the Supplementary Material together with an extra line of text. Food preferences $P_{ij}$ for predator of type i, and prey of type j are obtained from Bezhenar et al. (2016).
l. 284 "Table S9 in the Supplementary Material contains food preferences for organisms in the food web used in the Poseidon-R model."

(4) The modification of the dynamic radionuclide uptake model for strontium and caesium by salinity driven transfer parameters for the marine food web and its integration in POSEIDON-R was given in a paper by Heling and Bezhenar (2009). More data is necessary to verify temperature dependence of the MCKA parameters in further studies.

(5) A detailed comparison with analysis of the one-compartment model and MCKA model for FDNPP case study was added in the paper following suggestions by Reviewer #2. See answers on these comments. We did not compare MCKA and the one-compartment model with the single-feeding experiment because a one-compartment model cannot describe the fast transfer processes governed by eqn. (1).

[Figure]

**Figure 2** Schematic of the multi-compartment kinetic-allometric model.

**Specific comments**
**(1)** *In your introduction, you do not mention the POSEIDON-R model that you used to do the simulations of the accidental releases, you should at least present it. Without reading the*

*abstract, you do not expect to have a coupling of two models in the last part. More globally, I am not use to this type of presentation of an article, that is why I was a little bit confused reading the manuscript. I felt more like reading a report, even if I understand the importance of each part.*
**Answer.** We added text in the Introduction section:
l. 58 "The developed multi-compartment kinetic-allometric (MCKA) model was embedded into the box model POSEIDON-R (Lepicard et al., 2004; Maderich et al., 2014a,b; 2018b; Bezhenar et al., 2016), which describes transport of radionuclides in water, accumulation in the sediment, and transfer of radionuclides through the pelagic and benthic food webs."

**(2)** l. 48: *I am not a specialist of PBPK models but you wrote that fish PBPK models do not include scaling allometric relationship between metabolic rate and organism mass. However, the PBPK model of Grech et al. 2019 which takes into account the effect of growth on the cardiac output and oxygen consumption rate.*
**Answer.** Thank you for suggestion. We changed text accordingly:
l. 51 "Note that, with the exception of model (Grech et al. 2019), PBPK fish models do not include scaling (allometric) relationships between metabolic rates and organism mass."

**(3)** l. 107: *Could it have not be possible to adapt the dynamic budget theory (DEB) to model this? Could you explain better why this value of ¾ power? It is specific to fish?*
**Answer.** We plan to consider possible use of DEB in future modeling. As mentioned in l. 113 "we employed quarter-power scaling for uptake, elimination and growth rates derived from general theory (West et al., 1997)…" applicable for whole specter of organisms. An important consequence of this theory for the considered model is the equation (18) in which dependence on the body mass excluded. The text was changed accordingly:
l. 113 "Here, we employed quarter-power scaling for uptake, elimination and growth rates derived from general theory (West et al., 1997). This theory predicts for all organisms a 3/4 power law for metabolic rates. It describes transport of essential materials through space-filling fractal networks of branching tubes assuming that the energy dissipation is minimized and that the terminal branch of the network is a size-invariant. The scaling relations are…"

**(4)** l.115. *So, if I understand well, the structure of the model if generic for different fish species but the parameters values are specific to the species depending on their weight? Maybe you should mention it for more clearly. For two different fish species of the same weight, could you not have an inter-species variability of the model parameters?*
**Answer.** The text was modified as
l. 400 "The food and water uptake rates, elimination rate and growth rate depend on the metabolic rate, which is scaled by the fish' mass to the 3/4 power, but do not depend on the radionuclide. At the same time, the activity is distributed between the different tissues and organs according to the tissue assimilation efficiencies, which differ per radionuclide (Table 2), but that do not depend on fish mass. Therefore, the transfer rates can be associated with specific radionuclide and fish mass as shown e.g. in Tables S2-S4 and S6-S8. The position of the fish species in the trophic level also affects the concentration of activity in the organism."

**(5)** Fig 4. *Why is there no curve for dogfish? If you could not simulate BCF for dogfish, maybe you should withdraw the concerning data points.*
**Answer.** Calculated dogfish curve almost coincide with turbot curve. We restored it in figure.

**(6)** *Mathews et al. 2008 (l. 190) should not be Mathews and Fisher 2008?* (l. 208)
**Answer.** Done

**(7)** l. 242. *A graphical representation of the results of the sensitivity analysis in SI rather than a table would be of a great help to clarify the results. Why did not you make a global sensitivity analysis to better understand interactions between parameters?*
**Answer.** Thank you for suggestion. We replaced the table with a figure in Supplementary Material. In this paper, we limited ourselves to the One-At-a-Time method of relying on the results of previous studies (e.g. Bezhenar et al., 2016).

**(8)** l. 275. *A schematic representation of the one-compartment model would be required in SI as well as a table with the parameter values.*
**Answer.** The two parameters of one-compartment model are given and discussed in the main text of paper (lines 289-293 and 379).

**(9)** l. 276. *I do not understand the sentence.*
**Answer.** The text was changed accordingly. See also answer on comment 10.
l. 295 "The nested boxes (`inner' and `coastal' boxes) inside the regional box no. 68 in the Baltic Sea box system were added to resolve the radionuclide concentration in the near field (Fig. 6)."

**(10)** l. 277. *What are the "inner box" and "coastal box" exactly? I do not really understand as I am not familiar with the POSEIDON-R model. It should be better clarified.*
**Answer.** The inner and coastal boxes were explained as suggested:
l. 288 "The POSEIDON-R model can handle different types of radioactive releases: including atmospheric fallout and point sources associated with routine releases from nuclear facilities located directly on the coast or point sources associated with accidental releases (Lepicard et al., 2004). For coastal discharges occurring in the large ('regional') boxes, 'coastal' release boxes are nested into the 'regional' box system. The intermediary boxes between 'coastal' and 'regional' boxes are called 'inner' boxes."

**(11)** *Figures 7 and 8 could be coupled (6 panels).*
**Answer.** Thank you for suggestion. We merged figures 7 and 8 and 9 and 10.

**Technical corrections:**

*- Several spaces are missing:* l. 107, l. 241
**Answer.** Done

*- Error on the reference*: l. 278
**Answer.** Done